# A probabilistic map of emotional experiences during competitive social interactions

Joseph Heffner [1] & Oriel FeldmanHall [1,2 ✉]

Theories of emotion and decision-making argue that negative, high arousing emotions—such as anger—motivate competitive social choice (e.g., punishing and defecting). However, given the long-standing challenge of quantifying emotion and the narrow framework in which emotion is traditionally examined, it remains unclear which emotions are actually associated with motivating these types of choices. To address this gap, we combine machine learning algorithms with a measure of affect that is agnostic to any specific emotion label. The result is a probabilistic map of emotion that is used to classify the specific emotions experienced by participants in a variety of social interactions (Ultimatum Game, Prisoner's Dilemma, and Public Goods Game). Our results reveal that punitive and uncooperative choices are linked to a diverse array of negative, neutrally arousing emotions, such as sadness and disappointment, while only weakly linked to anger. These findings stand in contrast to the commonly held assumption that anger drives decisions to punish, defect, and freeride—thus, offering new insight into the role of emotion in motiving social choice.

[1] Department of Cognitive, Linguistic, Psychological Sciences, Brown University, Providence, RI, USA. [2] Carney Institute for Brain Science, Brown University, Providence, RI, USA. ✉email: oriel.feldmanhall@brown.edu

Although early models of decision-making largely ignored the influence of emotion[1], the last few decades have detailed the central role emotions play in guiding choices[2–5]. Emotions are particularly important for social cognition[6–8], where they are known to potently shape the decisions made during interactions with other people[9–11]. The coupling between emotion and social choice is perhaps most well-documented by research examining behaviors in competitive contexts, such as defecting during cooperative games and punishing norm transgressors. There is evidence that negatively arousing emotions—such as anger—act as the proximate mechanism motivating punishment[12–17]. For example, when reading about a moral violation[18–20] or receiving an unfair offer in an economic exchange[21], anger seems to scale with willingness to punish the transgressor. Physiological and neural data corroborate this account. Enhanced physiological arousal (i.e., skin conductance and pupil dilation) is observed when punishing those who behave unfairly and defecting after experiencing unreciprocated cooperation[22–24]. This heightened negative arousal is often taken as a proxy for anger[25,26], a narrative which has also been applied to the coupling between increased neural activity and decisions to punish and defect[27–29]. Although less prevalent, there is some evidence that other emotions such as sadness[30,31], disgust[15], and disappointment[32–36] also contribute to noncooperative behaviors. Taken together, the prevailing account is that strong, negatively arousing emotions play a critical role in discouraging prosocial decisions to reciprocate, cooperate, or help others[31,37].

However, the long-standing challenge of precisely quantifying nebulous emotional experiences[38,39] and the nature in which emotion is traditionally probed, leaves open the possibility that other emotions which do not neatly fit into this emotional taxonomy, might also shape social decision-making. For instance, interrogating how much anger a person feels in response to unfair treatment may artificially impose an expectation that the person ought to feel anger[40]. These types of directed probes also constrain the emotional experiences to a limited set chosen by the experimenter, often without allowing participants the option to report other emotional experiences. Moreover, attributing specific emotional states to increased physiological and neural activity may falsely lead to mis-identifying the emotion actually experienced[41]. Therefore, although emotion is often invoked as the lynchpin of motivated social decision-making[5,42], it remains unclear which emotions drive competitive decisions to punish and defect, and which emotions govern cooperative decisions to collaborate and reciprocate.

To circumvent these issues and precisely characterize the relationship between emotion and social choice, we developed a technique for understanding emotional experiences that combines participants' affective ratings of specific emotions with the affective experiences generated during social interactions. By leveraging both supervised and unsupervised learning algorithms trained to classify emotions, we let our data-driven framework reveal the nature of this relationship, effectively reverse engineering the emotions experienced during social interactions. This agnostic, unbiased approach identifies which specific emotional experiences are associated with competitive and cooperative choices, including punishment of a norm transgressor, defection after experiencing unreciprocated cooperation, and free riding at the expense of the common good.

In three experiments ($N = 1491$), we embed a mathematically tractable measurement of emotion into a series of economic games: The Ultimatum Game (Experiment 1), the Prisoner's Dilemma (Experiment 2), and a Public Goods Game (Experiment 3). This emotion measure partitions emotional experiences into a broad two-dimensional affect space of valence (pleasurableness, x-axis) and arousal (activation/intensity, y-axis)[43], parameterized by a 500 × 500 pixel grid (Fig. 1A). In our modified games, participants play as the second mover, reporting on this affect grid how they feel about the choices of the other player(s) (Fig. 1B). For example, if their partner proposes a highly unfair offer and the participant feels extremely negatively aroused, they could move their cursor to the upper left corner of the grid, demarcating their affective experience at a specific [x, y] location (participants were never asked to self-report a discrete emotion label during any of the economic games). Participants then respond by deciding to punish, defect, or free ride—depending on the game. Critically, before playing the economic game, all participants completed an emotion classification task in which they used their memory and prior knowledge[44] to place 20 labeled emotion terms (angry, surprised, happy, etc.) within the affect grid, such that each emotion is associated with a specific [x, y] coordinate (Fig. 1A). These [x, y] ratings of the labeled emotion terms are then used to train machine learning models to generate a representation of the group's "emotion map" within the affect grid space, which enable the model to predict which labeled emotion terms are most likely to be associated with a participant's unlabeled, affective experiences reported during the economic games.

To identify which specific emotions are associated with choices to punish, defect, and free ride, we trained three supervised machine learning classification algorithms (a neural network, k-nearest neighbors algorithm, and a support vector machine; Fig. 2), and one unsupervised algorithm (k-means clustering) on the valence and arousal ratings (i.e., [x, y] coordinates) provided by participants during the emotion classification task. Supervised models are evaluated on their cross-validation accuracy (see Methods), and once trained, the output of these models is the probability associated with an emotion's location in the affect grid at the population level (Fig. 2). Both the neural network (NN) and k-nearest neighbors (kNN) achieved high overall testing accuracy (NN: 35.80% and kNN: 35.97%, compared to null accuracy of 5%), while the SVM was a poor classifier (19.90%). While we chose the neural network model as our final model, results are comparable across NN and kNN. We applied the trained NN model to the unlabeled affective experiences reported during the economic games to infer an unbiased, participant-driven estimate of what emotion the person was likely feeling during the social interaction, without constraining participants' experiences (Fig. 1C).

## Results

**The emotions associated with decisions to punish.** Participants ($N = 715$) played a modified Ultimatum Game (UG), where the Proposer splits a sum of money with the participant, who can then decide to accept the offer, in which case the money is split as proposed, or reject the offer, in which case neither player receives any money (a classic form of costly punishment). Offers <20% of the total pie are typically rejected about half the time, which punishes the transgressor for behaving unfairly[45]. In our version, participants played as the Responder (or a third party, see Methods), and reported how they felt about the Proposer's offer on the affect grid before deciding to accept or reject. Before playing in the UG, participants rated 20 feeling terms in the emotion classification task. To gain insight into the emotions experienced during the UG, we combined the emotion classification data with the affect ratings made in all three experiments using machine learning algorithms. To accomplish this, we split the emotion classification data into a training (70%) and testing set (30%), and trained a feed-forward neural network (NN) on the training data. We used cross-validation to optimize the size of

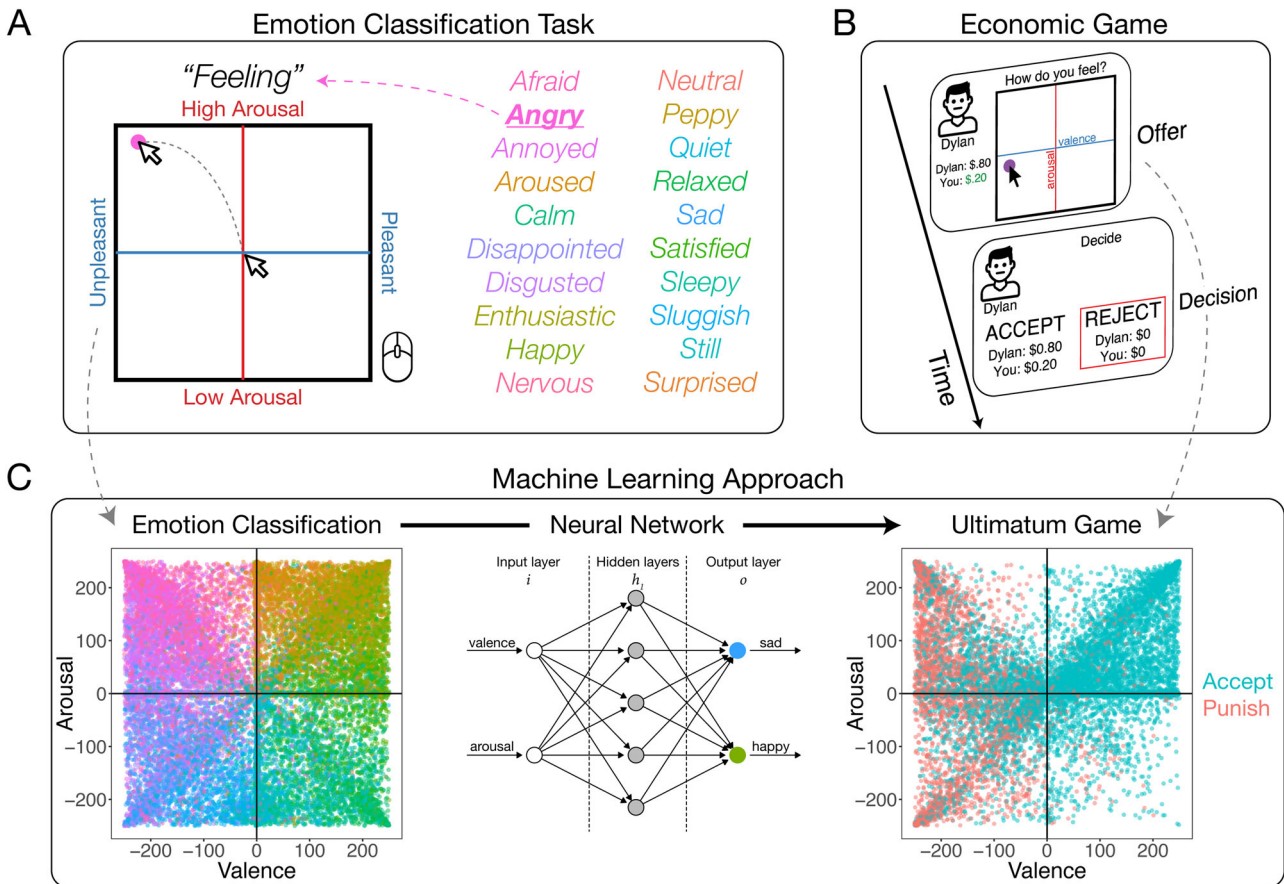

**Fig. 1 Task design. A** Emotion classification task schematic. Participants rated a variety of feeling labels using the arousal (vertical) and valence (horizontal) axes on a 500 × 500 pixel grid. The feeling labels for the emotion classification task are listed. **B** Economic game schematic. In a second task, participants played as the second mover in a dyad or group setting, deciding whether to punish a norm transgressor (UG) or defect in a cooperation game with one (PD) or three (PGG) partners. Here we provide a schematic of Experiment 1: The Ultimatum Game. Participants are paired with a partner, receive an offer, rate how they felt about that offer, and choose to accept or reject the offer. **C** Machine learning approach. Emotion classification ratings (colors correspond to emotion classification task words) were used to train a feed-forward neural network, shown in a simplified schematic predicting two emotion classes (see Methods). After cross-validation, the trained neutral network was applied to the unlabeled emotion experiences in the Ultimatum Game. Ultimatum Game data is color coded by the choice to accept (blue) or reject (red).

a single-hidden layer to prevent overfitting (see Methods). The testing set was used to validate the accuracy of the NN model by comparing the NN predicted emotion classifications, to the real emotion labels in the test data (NN achieved 35.80% accuracy, compared to a null accuracy of 5%). The resulting cross-validated NN model was applied to the unlabeled affective experiences reported in response to the Proposer's offer. Each affect rating was given a probability (or likelihood) of being classified as one of the 20 discrete labeled emotion terms (where all probabilities add up to 1). Because each affect rating was associated with a choice to accept or reject, results were averaged within participants and then across choices to gain insight into which emotions are linked to decisions to punish.

**The 2D structure of emotion.** Results from the emotion classification task reveal at the population level how emotions are organized in the two-dimensional valence-arousal space. For example, the contour plots of the 2D density distributions for the emotion anger reveals a tight, densely centered set of responses that fall in the high arousal, unpleasant grid space, an experience that might be described as "rage" (Fig. 2A). There is also, however, a small dense cluster of responses that are neutrally arousing and highly unpleasant, an experience probably more akin to 'quiet anger'. These two distinct clusters of responses suggest that

at the population level, there is a qualitative difference in the emotional experience labeled anger. Other emotions, such as disappointment, appear to have greater heterogeneity, reflected by responses spanning a greater swath of the affect grid, especially along the arousal dimension. This suggests that there is no single common emotional experience that neatly describes the emotion called disappointment, and even basic emotions such as anger appear to have affective variability (Fig. 2B). The heterogeneity in the structure of emotions can be tested by quantifying the distribution (e.g., standard deviation or interquartile range)[46,47] and comparing the variance of each emotion[48]. For example, many negative emotions (i.e., afraid, angry, annoyed, disgust, disappointment, and sadness) have comparable variances along the valence dimension, but most emotions' variances differ along the arousal dimension (see Supplementary Fig. 3). The distributions themselves can also be formally compared by determining whether a specific distribution is unimodal (e.g., Hartigan's dip statistic)[49,50]—which, depending on the peaked-ness of the distribution, might suggest a common emotional experience at the population level, or multimodal—which would suggest greater heterogeneity at the population level (see Supplementary Tables 1, 2). Together, these techniques help to reveal the structure of emotion and offer insight into the heterogeneity of the human emotional experience.

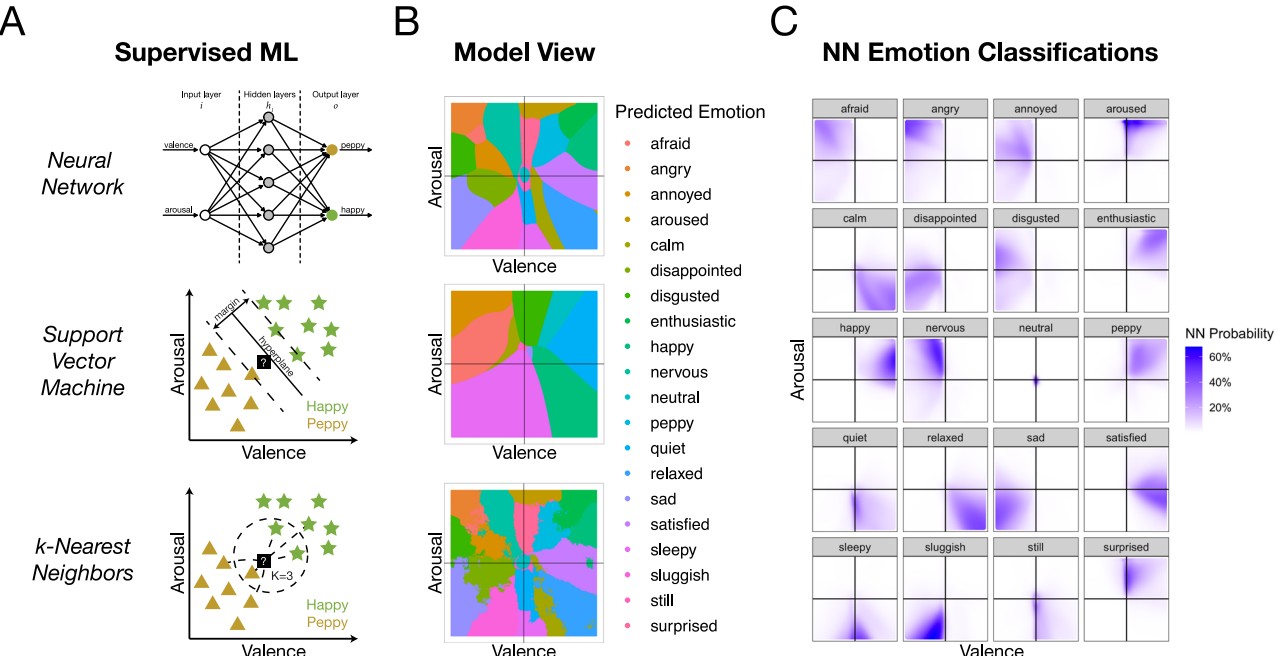

**Fig. 2 Machine learning approaches. A** Supervised machine learning. Three supervised machine learning algorithms were used to train a classifier on all 20 emotions from the emotion classification data. Simplified schematics for each approach are shown demonstrating how the model classifies toy data as either "happy" or "peppy". Model-specific parameters were optimized using tenfold cross-validation on the training dataset. **B** Model view. Graphs show most likely emotion class for each valence-arousal location which visualizes one model's view of the emotion space. **C** Neural network emotion classifications. Visualizes the probability of each emotion label classification for the emotion space, with darker colors indicating higher probability.

**Sadness and disappointment are associated with punishment.** The neural network trained on the emotion classification data predicts the likelihood a given emotion is experienced after receiving an offer in the UG. Contrary to popular emotion-punishment theories, our results reveal that the top three emotions associated with decisions to punish are sadness (13.47%), disappointment (12.82%), and disgust (12.54%), with anger (10.28%) identified as the 5th most likely emotion to be experienced after receiving an offer (Fig. 3A). Paired $t$ tests reveal, sadness ($t$ (558) = 4.65, $p < 0.001$, $d = 0.20$), disappointment $t$ (558) = 4.06, $p < 0.001$, $d = 0.17$, and disgust ($t$ (558) = 5.52, $p < 0.001$, $d = 0.23$) are all significantly more likely to be associated with punishing compared to anger. When accepting the offer, the top three emotions most likely to be experienced are satisfied, happy, and peppy (Fig. 4A). To assess the degree to which a given emotion is specifically associated with a decision to punish compared to accept, we can contrast the model likelihoods for each choice (e.g., a delta measure computing the likelihood of feeling anger when punishing —likelihood of feeling anger when accepting). Higher values indicate that an emotion has a stronger association with punishment, and lower values indicate that an emotion has a stronger association with acceptance (Fig. 4B). Emotions such as sadness, disgust, anger and disappointment were all more intimately linked to decisions to punish, compared to when deciding to accept.

**Negatively valenced, high arousal emotions less likely to predict punishment.** To test the more general notion that emotions which are negatively valenced and highly arousing are commonly linked to punishment[14,28], we used an unsupervised machine learning approach called k-means clustering. The k-means procedure ignores the emotion labels from the emotion classification task and partitions the emotion space into nine equally sized clusters (we chose $k = 9$ because it forms a three-by-three checkered emotion space; Fig. 5A). These nine clusters are roughly equivalent to delineating all possible combinations of low, medium, and high

levels of arousal and valence. The objective of the k-means algorithm is simple: While ignoring actual emotion labels, take the population-level responses from the emotion classification task to discover any underlying patterns for how the group represents each emotion (see Methods). This approach allows anger-punishment theories a fighting chance by interrogating—in an even more unbiased approach—the hypothesis that negatively valenced, highly arousing emotions (such as, anger) predict decisions to punish.

Results show that cluster 3, which corresponds to high negative valence and neutral arousal, was the most frequent affective experience associated with decisions to punish—almost twice as frequent ($\chi^2(1) = 249.68, p < 0.001$) as the next predictive cluster (cluster 1: high negative valence and high arousal; Fig. 4B: note that the proportion of punish and accept choices which naturally fall into these clusters are not uniform and as such, the base rates of punishment in each cluster differ, e.g., out of all choices to punish, 11.5% fall within Cluster 3 while only 3.44% fall within Cluster 1; see Supplementary Fig. 7). Examining which emotions fall into cluster 3 reveals disappointment as the most representative emotion term, whereas anger is most representative of cluster 1 (Fig. 4A). When deciding to accept the offer, cluster 5 (neutral valence and arousal; represented by the emotion term neutral) was almost twice as likely ($\chi^2(1) = 520.22, p < 0.001$) as cluster 2 (neutral arousal and positive valence; represented by the emotion term satisfied). Given that we still find that the affect cluster associated with anger (cluster 1) is not representative of decisions to punish (even when accounting for base rates) suggests that our results cannot simply be explained by how people interpret specific emotion terms. Instead, these results provide converging evidence that negative valence and high arousal affective states are not the predominate motivator of punishment.

**Highly unfair offers increase coupling between negatively valenced, neutrally arousing emotions and punishment.** Being treated unfairly is a strong and consistent predictor of decisions to

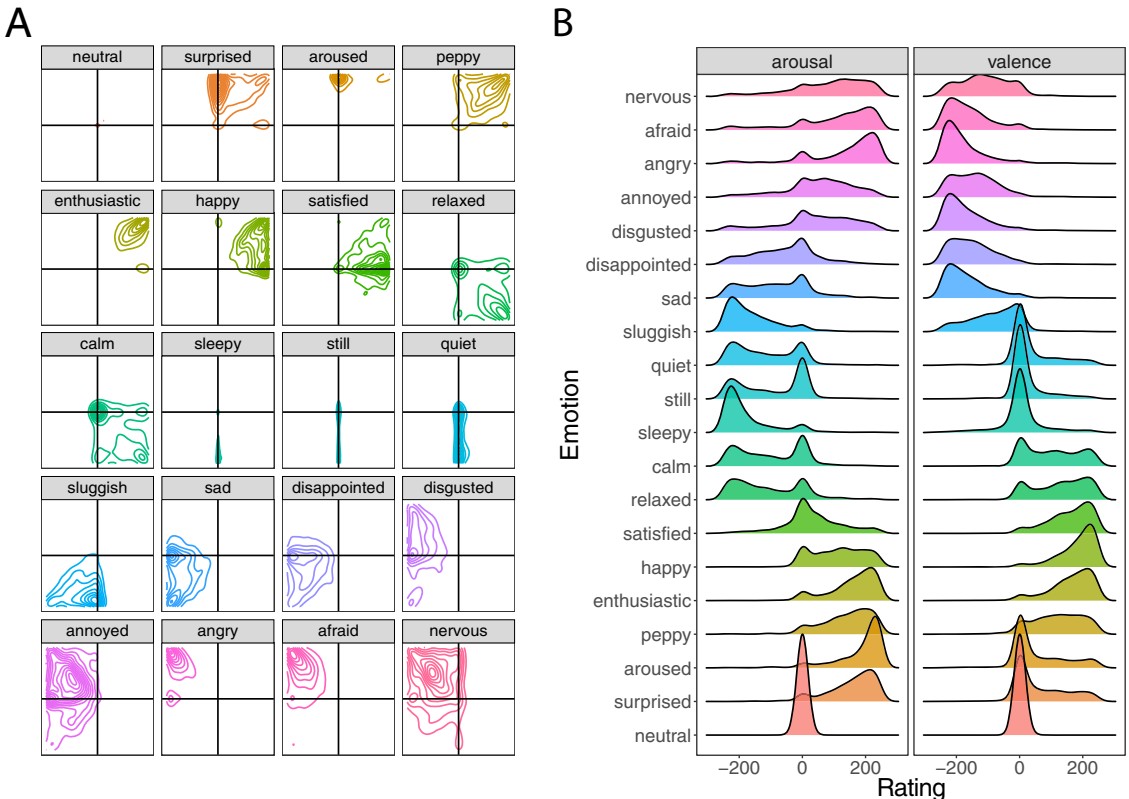

**Fig. 3 Emotion classification plots. A** Two-dimensional density plots of where participants (N = 1491) placed each emotion term in the emotion classification task. The x-axis represents valence and the y-axis represents arousal. Contour lines illustrate different levels of density at the population level, see Supplementary Information for more detailed visualizations. **B** One-dimensional density plots illustrating the ratings of emotion terms in the emotion classification task, plotted separately for valence and arousal.

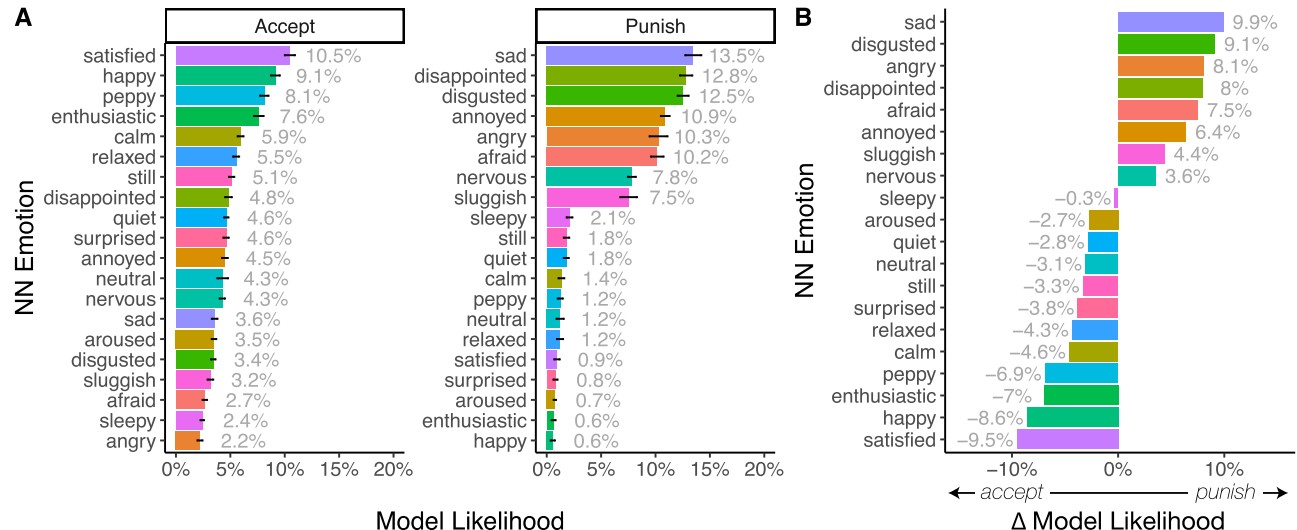

**Fig. 4 Neural network emotion classifications. A** Classifications by decisions to accept and punish. The neural network model trained on the emotion classification data is applied to the unlabeled emotion ratings from the Ultimatum Game. Each data point is assigned a probability of each emotion class and model likelihoods are averaged within participant (N = 715) and then across choice. Bars represent mean values while error bars reflect 95% CIs. **B** Difference in probability of choice given an emotion. Model likelihoods are averaged within participants, then across choices to reveal which emotions are more likely to be associated with decisions to accept and punish. For example, the emotion sad is more specific to decisions to punish than accept, whereas the emotion sleepy is equally associated with both choices.

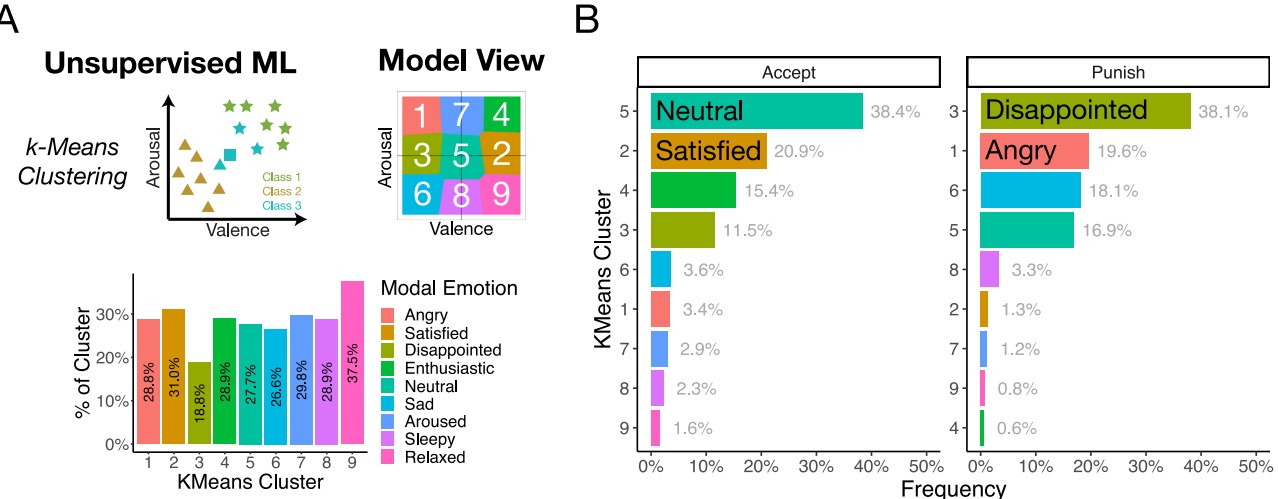

**Fig. 5 k-means clustering approach. A** k-means cluster schematic and interpretation. We specified the k-means algorithm to identify nine clusters which roughly forms a three-by-three grid varying in low, medium, and high valence and arousal. Each cluster is numbered and the frequency of the modal emotion for each cluster is shown to illustrate each space of the emotion grid. **B** Classifications by decisions to accept and punish. The k-means cluster model was applied to the unlabeled emotion ratings from the Ultimatum Game. Each data point is categorized by a single cluster and the frequency of each cluster is shown for accept and punish decisions.

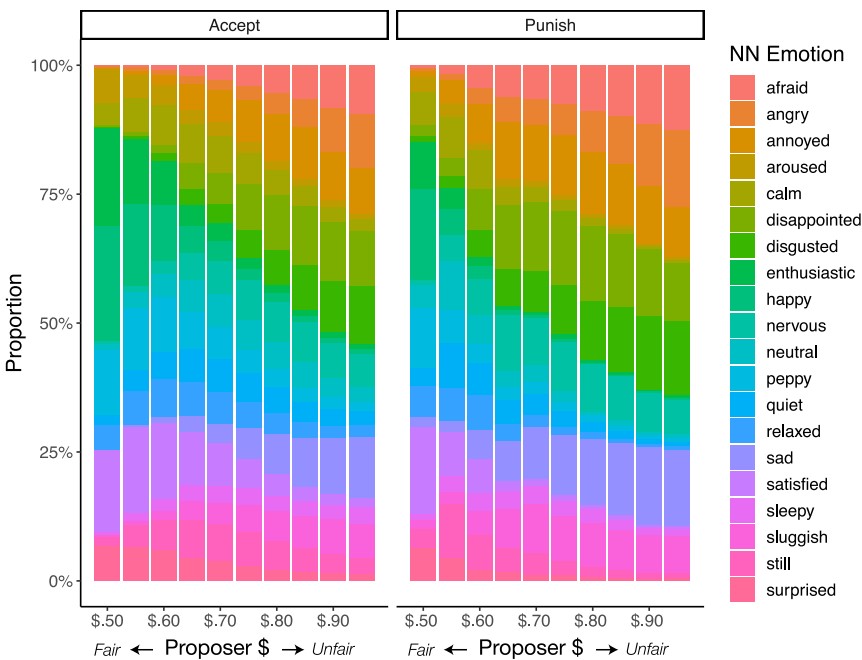

**Fig. 6 Neural network classifications across unfairness levels.** Unfairness indexes the amount of money kept by the Proposer (out of $1) and ranges from fair ($0.50, $0.50) to highly unfair offers ($0.95, $0.05). The proportion refers to the model likelihood of each emotion class within participants ($N = 715$), and then across choice (accept or punish) and each unfairness level.

punish, especially in the UG[51]. The association between unfairness and punishment provides an additional, strong test of which specific emotional experiences evoked by being treated unfairly leads to punishing. Using the emotion classification likelihoods derived from the neural network, we calculated the average probability of each emotion class at every level of unfairness, given a decision to punish or accept the offer. As offers become more unfair, people experience more negative emotions including sadness, disgust, disappointment, and anger (Fig. 6). The model identified disappointment as the most likely emotion for most offer types. Only after experiencing the most unfair offer (5% of the total pie), does anger become the top emotion linked to punishment, followed by sadness and disgust (paired sample t test

comparing anger with sadness $t$ (537) = 0.03, $p = 0.98$, $d = 0.001$) and disgust ($t$ (537) = 0.59, $p = 0.55$, $d = 0.03$).

**Individual-specific emotion representations**. While the machine learning approach combines the data across all participants to classify affective experiences as a population-level estimate, we can also tailor our analysis to each individual's unique affective experience. To do this, we directly tied a participant's specific emotion classification responses to their reported affective experiences in the UG, which creates an individual-level estimate. For example, if a particular participant rates anger as being a negative, low arousal state (i.e., a "quiet" anger), then this analysis

would be sensitive to that participant's idiosyncratic representation of anger, which might deviate from the group's representation. To create an individual-level estimate for the emotions felt during decisions to punish, we computed the inverse Euclidean distance between a participant's affect ratings made during the UG and the coordinates reported for each emotion term in the emotion classification task (see Methods for how we converted distance to probabilities). Essentially, this allows us to capture the probability that a person's unlabeled affect ratings made during a social interaction are similar (or not) to their unique experience of a given emotion term (e.g., anger). Results reveal that at the individual level, disgust (9.41%), disappointment (8.15%), and anger (7.97%) were the most frequently experienced emotions (disgust was significantly more likely to be experienced than anger ($t(558) = 3.50$, $p < 0.001$, $d = 0.15$), while disappointment was not ($t(558) = 0.42$, $p = 0.68$, $d = 0.02$); see Supplementary Fig. 9). When deciding to accept, the three emotions most likely to be experienced were neutral, happy, and satisfied. These individual-level estimates, which account for the presence of idiosyncratic emotion representations, largely align with the population-level results.

**The emotions associated with decisions to defect**. By using machine learning algorithms to infer the emotions experienced during a social exchange, Experiment 1 found that disappointment is the most representative emotion of decisions to punish after being treated unfairly, whereas anger seems to play a much smaller role than originally believed. In Experiment 2 we wanted to further test the link between anger and social decisions in a different competitive context. Thus, we leveraged a similar experimental framework, but this time examined choices to cooperate or defect. In Experiment 2 ($N = 306$), participants played a modified sequential Prisoners' Dilemma[52], where players decide whether to cooperate or defect with one another. In our version, both players are given $1 and asked how much they want to contribute to a joint outcome. Any amount contributed is multiplied by 1.5 and redistributed evenly amongst both players, creating a continuous version of cooperate (contribute $1) and defect (contribute $0). Participants report how they feel about their partner's contribution (which ranged from $0 to $1, see Methods) on the affect grid before deciding how much they themselves want to contribute.

**Sadness and disappointment are associated with defection**. To make the analysis analogous to Experiment 1, we binned continuous contributions into decisions to defect ($0–$0.49) and cooperate ($0.50–$1). Results reveal that disappointment (8.83%), sadness (8.72%), and disgust (7.38%; in that order) were the most likely emotions associated with defection, all of which were more likely than anger (5.11%)—which ranked as the 9th most likely emotion. Paired $t$ tests reveal that disappointment ($t(278) = 8.26$, $p < 0.001$, $d = 0.50$), sadness ($t(278) = 6.84$, $p < 0.001$, $d = 0.41$), and disgust ($t(278) = 8.37$, $p < 0.001$, $d = 0.50$), are all significantly more likely to be associated with defection in the PD compared to anger. These results hold if we analyze choice continuously ($0–$1), which only further confirms that anger ranks significantly below (8th) both sadness (1st) and disappointment (2nd; see Supplementary Figs. 11, 12). In contrast, decisions to cooperate are linked to happiness (16.32%), satisfaction (14.60%), and enthusiasm (13.88%; in that order; see Supplementary Fig. 10 for full ranking). Results from the unsupervised machine learning approach which makes no assumptions about specific emotion labels per se, illustrates that the main cluster associated with defection is cluster 5 (i.e., feelings of neutral valence and neutral arousal)—which is more than five

times more common than cluster 1 (i.e., feelings of high negative valence and high arousal; $\chi^2(1) = 662.88$, $p < 0.001$). The individual-level analysis dovetailed with these results, demonstrating that anger ranks as the 11th most likely emotion associated with decisions to defect (see Supplementary Fig. 14 for full ranking). Together, these results provide converging evidence that anger is highly unlikely to be the main emotion motivating competitive social decision-making.

**The emotions associated with decisions to free ride**. To further test the generalizability of these results, in Experiment 3 ($N = 470$), participants played a modified sequential Public Goods Game[53], where participants and three players collectively decide how much to contribute to a common pool. All players were given $1 and told that any amount contributed would be multiplied by two and then redistributed evenly amongst all players. Participants reported how they felt about their partners' collective contribution on the affect grid before determining their own contribution (see Methods).

**Sadness and disappointment are associated with free riding**. As in Experiment 2, we binarized continuous contributions into decisions to defect ($0–$0.49) or cooperate ($0.50–$1). Results reveal the top three emotions associated with defection in a group context are, sadness (10.36%), disappointment (9.63%), and sluggishness (8.38%) —with anger (4.74%) ranking 8th (see Supplementary Fig. 15 for full ranking). Paired $t$ tests revealed that sadness ($t(442) = 12.9$, $p < 0.001$, $d = 0.61$), disappointment ($t(442) = 14.0$, $p < 0.001$, $d = 0.67$), and sluggishness ($t(442) = 7.60$, $p < 0.001$, $d = 0.36$) were all significantly more likely to be associated with decisions to free ride than anger. As in Experiment 2, these results hold if we analyze choice continuously ($0–$1), illustrating that both sadness (1st) and disappointment (2nd) rank significantly above anger (7th; see Supplementary Figs. 16, 17). As before, decisions to cooperate were linked to feeling happy (13.47%), enthusiastic (12.95%), and satisfied (12.06%). The unsupervised machine learning model results showed the main cluster associated with defection was again cluster 5 (associated with neutral valence and arousal), which was more than six times more likely than cluster 1 (associated with high negative valence and arousal; $\chi^2(1) = 4948.80$, $p < 0.001$). Moreover, the individual-level analysis revealed anger as the 9th most likely emotion when deciding to free ride (see Supplementary Fig. 19).

## Discussion

Which emotions best predict decisions to punish, defect, and free ride in competitive social interactions? While prior research argues that feelings of anger play a predominant role in motivating such competitive social choices[14], our data-driven machine learning approach finds that other emotions, primarily disappointment and sadness, are far more likely to be associated with punishing, defecting and free riding. Despite the intuitive appeal of negative, highly arousing emotions being linked to competitive social decision-making, we did not find good evidence that this was the case—regardless of the number of people involved or the unique social tensions of the interaction. When we account for individual variability in the representation of emotion, we find that disappointment is still more frequently experienced than anger. Even when ignoring emotion labels entirely and focusing exclusively on unlabeled affect responses, we find little evidence that highly arousing and negatively valenced feelings, an affective experience traditionally associated with the emotion anger, shape these decisions. Instead, negatively valenced and neutrally arousing emotions—experiences akin to sadness or disappointment—are the most common affective experiences associated with punishing, defecting, and free riding.

By not forcing people to report emotions assumed to be involved with certain social choices, we observed a diverse array of emotional experiences driving competitive social decision-making. Although a forced-choice design can at times be a useful tool, it necessitates that participants' responses align with the expectations of the experimenters[54], and it may inflate the importance of certain emotions according to social norm theory (e.g., "I ought to feel angry after being treated unfairly")[55,56]. By using unlabeled affect measurements to probabilistically classify which emotional experiences relate to discrete social choices, our framework sidesteps these issues. Moreover, this unbiased, data-driven approach reveals how unlabeled affective responses at the individual level scaffold the structure of a given emotion at the population level. For example, some people might be motivated by "quiet anger" when punishing, while others are more motivated by an "intense rage". Since no singular approach can accurately describe the subjective emotional experience of all individuals[57], future research should consider this type of emotion measurement in conjunction with other techniques, such as self-report or physiological measures.

Detailing how emotion interacts with decision-making is foundational for understanding the mechanisms guiding social cognition. As one prominent example, it has long been theorized —and demonstrated—that specific emotions motivate certain actions, such as a fear response leading to a tendency to flee[2,58,59]. The framework employed by traditional affect theories and emotion measurements suggests a one-to-one mapping between certain emotions and choice. Yet, leveraging a broader data-driven approach that relies less on the discretization of emotion and which makes no assumptions on the likelihood of candidate emotions experienced, enabled us to discover the wide heterogeneity of the feelings evoked during a specific set of social interactions. That is, the emotions driving punishment, defection, and free riding are far more diverse than previously thought.

Given that we found highly negatively valenced emotions to be more closely linked with decisions to punish, defect, and free ride, our results highlight the essential role of valence in predicting social decisions. In contrast, high arousal was not as readily linked with these choices. This diverges from prior work implicating autonomic nervous system arousal as a key component motivating decisions to punish[25,28], which suggests there might be a difference in people's awareness of their emotional processes and how the body physiologically indexes that process[60]. It may be the case that people are less able to appraise their body's physiological arousal states, which would indicate that self-reported arousal does not bear a one-to-one mapping with physiological arousal. Instead, there may be different affective roles between the conscious level associated with reporting arousal states, and the more implicit measurements garnered at the physiological level. Indeed, the relationship between physiology and subjective evaluation of emotions is not straightforward[61], as physiological arousal can be interpreted in different ways depending both on the context[62], and an individual's level of introspection[63]. Further research in this area, particularly efforts to characterize the relationship between physiological and reported arousal[64], can help better characterize this interactive, dynamic process, and how it might bias choice.

Although the two-dimensional affect structure of valence and arousal is central to theories of emotion[56,65–67] and grounded in the body's neurobiological system[68], some aspects of emotional experience may not be properly captured by a two-dimensional valence-arousal model. For example, anger and disgust are both associated with unpleasant, high arousal affect, but are conceptually linked to different behavioral responses: Anger is typically associated with approach-motivated responses and disgust with avoidance-motivated responses[69–72]. While the

two-dimensional valence-arousal space offers a quick measurement of affect and is thus useful for fast, repeated decision-making paradigms, adding in more dimensions (e.g., approach/avoid, anticipated effort, control)[73–75], could be helpful in better characterizing the structure of the emotional space, and should improve classification accuracy by separating similar emotions along other dimensions.

Although the general link between emotion and choice is established in the field of human social cognition, less is understood about the relationship between the specific emotions guiding social choices. Our results suggest that framing anger as a motivating force may mischaracterize the nature of the relationship. While we observed large variability in the emotions driving competitive social choices, emotions characterized by high, negatively valenced, but muted arousal appeared to play the most central role in driving decisions to punish, defect, and free ride. By combining the continuous, dimensional structure of affect with discrete emotion states, our novel analytic approach allowed us to identify the likelihood that a particular emotion drives a social choice, while also revealing the heterogeneity of emotions experienced during these interactions. As the disciplines of emotion and behavioral economics advance, we can build on this progress to further characterize how emotions guide decisions in a variety of contexts.

## Methods

**Participants.** Across three experiments, participants ($N = 1820$) completed an emotion classification task followed by one of three economic games: An Ultimatum Game (UG; Experiment 1 $N = 906$), a Prisoners' Dilemma (PD; Experiment 2 $N = 395$); or a four-person Public Goods Game (PGG; Experiment 3 $N = 519$). Using preregistered criteria based on existing work[76], we excluded participants (N=329 total; UG exclusion = 191; PD exclusion = 89; PGG exclusion = 49) who rated neutral outside of a $100 \times 100$-pixel square in the center (i.e., participants were instructed to rate "neutral" in the center of the dARM). The final sample for the UG was $N = 715$ (320 Females, mean age = $34.4 \pm 10.1$), the final sample for the PD was $N = 306$ (131 Females, mean age = $35.5 \pm 11.2$), and the final sample for the PGG was $N = 470$ (238 Females, mean age = $33.0 \pm 10.5$). Thus, the final sample across all three experiments was $N = 1491$ (689 Females, mean age = $34.2 \pm 10.5$). Participants were recruited from Amazon Mechanical Turk and received monetary compensation and provided informed consent in a manner approved by Brown University's Institutional Review Board under protocol 1607001555.

**Procedure.** In the Emotion Classification Task, participants rated a series of feeling words on an affect grid varying on valence and arousal (Fig. 1), modified from the traditional affect grid[77–79]. A person who is rating anger might, for example, place their cursor at the top left corner of the grid, indicating high arousal and negative valence (participants are free to report any subjective interpretation of anger, such as, a quiet anger). The 20 feeling terms (neutral, surprised, aroused, peppy, enthusiastic, happy, satisfied, relaxed, calm, sleepy, still, quiet, sluggish, sad, disappointed, disgusted, annoyed, angry, afraid, nervous) were selected from past research to represent the octants of the emotion circumplex space evenly[44] or because they have been implicated in decisions to punish and defect during social interactions[80]. Participants then completed one of three economic games which were structured in a similar way. Finally, participants completed a series of individual difference questionnaires that were not analyzed in this research.

*Ultimatum game.* Participants completed 20 one-shot Ultimatum Games as either the Responder ($N = 543$) or a third-party ($N = 172$). Since Responders and third-parties' affective responses were not significantly different, we collapsed across role (see Supplementary Information). After receiving the offer, participants reported how they felt on the affect grid before deciding to accept or reject. Participants received an even distribution of offers ranging from fair ($.50, $.50) to highly unfair ($.95, $.05). Unfairness was operationalized numerically according to the amount of money kept by the partner. Participants engaged with new partners on each round. The data from the UG was part of a dataset collected for a previously published study[76].

*Prisoners' dilemma.* Participants were paired with a new partner (denoted by a face and name) on each round. Both players were given $1, which they could use to contribute to a collective pot. Any amount contributed was multiplied by 1.5 and redistributed evenly between the pair. The tension lies between making more money by defecting at the expense of the other player's monetary gain. Our PD version was structured in a sequential manner, so that emotion could be measured

during the game, and to compare it to the Ultimatum Game data. Participants decided how much they wanted to contribute in $0.10 increments. Participants completed 22 one-shot Prisoners' Dilemma rounds, where partners contributions ranged from defection ($0) to full ($1) cooperation.

*Public goods game.* The Public Goods Game was similar to Experiment 2 except participants were paired with three other players (players were denoted by an anonymous Amazon Mechanical Turk IDs). The players were each given $1, which they could use to contribute to the common pot. Once contributed, the money was doubled and redistributed evenly between all four players. A player could make more money at the expense of the outcomes of others by free riding while the others pay into the common pot. After participants were informed of the collective amount contributed by the other players, they reported how they felt on the affect grid about the collective contributions. Then they decided how much to contribute themselves (up to $1, in $0.10 increments). Participants completed 62 sequential one-shot Public Goods Games with an even distribution of partner contributions ranging from none ($0) to full ($3 total contribution).

**Machine learning models**. In the Emotion Classification Task participants rated a variety of emotion words on the $500 \times 500$ affect grid. Participants rated 20 emotions, not all of which were related to punishing/accepting an unfair offer or free riding amongst a group (e.g., "peppy"). This data was used to develop the supervised machine learning classifiers. We trained three supervised machine learning classification algorithms (Fig. 2) using tenfold cross-validation on a 70–30 train-test split of the emotion classification task, and one unsupervised machine learning algorithm. Within the training data, we use tenfold cross-validation which randomly splits the training data into ten subsets. One subset is reversed for validation while the model is trained on the other subsets according to the model's specific algorithm. The model is tested on the reserved subset and an accuracy score is recorded. In this classification problem, accuracy is the proportion of correct classifications over the total classifications. A null accuracy would be 5% because this represents a model which simply suggests at each classification. Because the emotion classification dataset is balanced, meaning each of the emotion classes is equally represented, accuracy is a good metric for evaluating the model. The process is repeated until each of the ten subsets have served as the validation set and the cross-validation accuracy is the average of these recorded accuracies.

*Neural network.* Neural networks are a nonlinear statistical model and a popular choice for complex classification problems. We used a feed-forward single-hidden-layer neural network[81], which had two input nodes for valence and arousal, a single-hidden layer with a variable number of hidden nodes, and 20 output nodes representing the 20 emotions from the classification task. The number of nodes in the hidden layer was determined by cross-validation and the final model contained 27 nodes in the hidden layer, with a decay of 0.035, which were fully connected to the input and output nodes.

*Support vector machine.* Although support vector machines are typically used for binary classification, they can be a useful tool for multi-class classifications. The support vector classifier constructs a linear boundary in a large, transformed version of the feature space (in this case, valence and arousal ratings) by defining an optimal hyperplane that separates two classes. For our multi-class problem, we used a "one-against-one" approach or pairwise classification method which constructs $\binom{k}{2}$ classifiers where each one is trained on data from two classes[82].

Because our classes are not perfectly separable (i.e., they overlap and misclassifications exist within a margin of the hyperplane), we use cross-validation to determine the correct "Cost" (C) parameter, which defines the weight of how much of the data inside the margin of error contributes to the overall error. The final SVM model used a cost parameter of $C = 0.01$.

*k-nearest neighbors.* Another popular method for classification of data with a low number of dimensions is the K-nearest neighbors (KNN) classifier. The KNN classifier is a non-parametric approach that computes the conditional probability of the data class for any data point. This is done by comparing a data point to the class of data points in close proximity. The classifier chooses a neighborhood size, represented by the parameter $K$, and estimates the conditional probability for emotion cluster $j$ as the proportion of data in the neighborhood set, whose class is also $j$. The $K$ parameter and neighborhood size was determined through cross-validation, and the final KNN model used a neighborhood size of $k = 175$.

*Model selection.* Both the NN and KNN reached similar levels of overall testing accuracy (NN: 35.80%, KNN: 35.97%) and kappa (NN: 32.42, KNN: 32.60), while the SVM fit relatively poorly (SVM accuracy: 19.90%, SVM kappa: 15.68). While the KNN had marginally more accurate classifications, it also had undesirable properties, such as sporadic emotion islands (e.g., a tiny, isolated pocket of an emotion surrounded by a larger emotion cluster). The NN model achieved similar accuracy and had smooth emotion boundaries between classifications. For these reasons, we selected NN as the final model, but note that both models had comparable results. After training the NN model and validating its accuracy through

cross-validation, we applied the trained neural network to the unlabeled emotion data generated from the affect grid in the Ultimatum Game, Prisoners' Dilemma, and Public Goods Game. We generated a likelihood of this emotion data being classified in each of the 20 emotion categories. That is, each emotion rating across all trials of the behavioral economic games is given a probability for each of the 20 possible emotion classes, which together add up to 1. This approach allows us to identify which emotion states are likely to drive punishment, without potentially biasing the result in favor of any specific emotion (e.g., specifically asking "how angry do you feel?").

*K-means clustering.* K-means clustering is an unsupervised machine learning method for finding clusters in a set of unlabeled data. Although the emotion classification data was labeled, we can strip each emotion from its label for the purposes of finding optimal emotion clusters which vary on valence and arousal intensity. Given a desired number of cluster centers, the k-means procedure randomly defines the initial center of the cluster and then iteratively moves the center of that cluster to minimize the total within-cluster variance[83]. We chose nine clusters because it formed a rough three-by-three checkered emotion space representing high, medium, and low valence and arousal combinations.

*Euclidean distance measurements.* Although the Euclidean distance measurement used in the individual difference analyses are not formal machine learning models, they represent another way to classify emotions that are unique to each participant. We calculate the Euclidean distance between the 20 emotion terms in the emotion classification task and the unlabeled affective reports during each trial of the economic games. We convert Euclidean distance into a numeric probability using inverse distance weighting, where the probability is the inverse of the Euclidean distance for a given emotion over the sum of inverse Euclidean distances of all emotions. Smaller Euclidean distances indicate a higher probability that the unlabeled affective experience is similar to the valence and arousal rating of the emotion term (e.g., anger, disgust, etc.) for each participant separately.

**Reporting summary**. Further information on research design is available in the Nature Research Reporting Summary linked to this article.

## Data availability

Experimental materials information and all experiment de-identified data are publicly available at https://github.com/jpheffne/NC_emotion_classify. The materials used in this study are widely available.

## Code availability

Data analysis script notebooks are publicly available at https://github.com/jpheffne/NC_emotion_classify.

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

## Acknowledgements

Part of this research was conducted using computational resources and services at the Center for Computation and Visualization, Brown University. This material is based upon work supported by a Graduate Award in Brain Science from the Carney Institute for Brain Science (J.H.), and a Center of Biological Research Excellence P20GM103645 from the National institute of General Medical Sciences (O.F.H.).

## Author contributions

J.H. and O.F.H. contributed to designing the research, analyzing the data, and writing the paper.

## Competing interests

The authors declare no competing interests.
