## [Peer Review File · Nature Communications]

A probabilistic map of emotional experiences during
competitive social interactionsREVIEWER COMMENTS

Reviewer #1 (Remarks to the Author):

Positive comments

This paper takes a fresh and innovative look at how self-reported emotion categories map onto social decision making, and in particular punishment of others during social interactions. The main thrust of the paper seems to challenge the link between anger and social punishment (rejecting unfair offers in the Ultimatum Game, defection in Prisoner's Dilemma and Public Goods games), by claiming that a broader set of emotions (e.g., nervous, afraid, or even sluggish) are also associated with punishment. This is used to argue that anger is less important as a motivator of punishment than researchers previously thought.

The concept is clear and the paper blends emotion research with machine learning models in an accessible way. The figures are informative and aesthetically beautiful. (But: There are very few figures showing any actual study data or inferential statistics, which is an important limitation for me).

The conclusions are surprising and would be interesting if borne out: e.g. "negative valence and high arousal affective states are unlikely to motivate punishment". But (unfortunately) I'm not convinced that they're warranted based on the present manuscript.

Potential weaknesses

Theoretical claims. This paper seems to imply that because people punish when they're angry, the literature suggests that people reject offers only when they're angry. The paper challenges the latter claim, but not the former. I'm not sure that many emotion theorists would endorse the latter. For example, with many people really endorse the idea that people reject an unfair offer when they are angered by it but do not reject disappointed or saddened by it?

Put another way, I can imagine many situations in which people are disappointed or disgusted by propositions and not truly "angered", but still reject those propositions. Some such propositions were studied here – e.g., being offered \$3 out of a possible \$10 may be "disappointing", and I may reject the proposal. $P(\text{reject} \mid \text{disappointment})$ may be high. But this does not inform on whether $P(\text{reject} \mid \text{anger})$ is high. It does inform on the reverse inference, $P(\text{anger} \mid \text{reject})$, i.e., that I reject only when angry. But I do not believe the literature makes any strong claim about this hypothesis, and high $P(\text{anger} \mid \text{reject})$ is not very plausible in the first place.

Alternative models. The paper maps rejection of offers (social punishment) onto a relatively complex space of emotions. The implication is that these emotions are important, even potentially as causal factors that impact decision-making. But what if the decision space is much simpler? For example, responders in the ultimatum game might choose to reject offers based on a policy decision, if they are below 20% of the total pie, for example. This is an example of a much simpler model. Does a policy model explain the data better than a model that requires knowing whether one is sad, disappointed, angry, or sluggish?

Likewise, is position in valence/arousal space a stronger predictor of accept/reject decisions than the emotion category labels? Does the reported category add predictive information, or is it superfluous? I suspect that in a logistic (or nonparametric logistic) regression with valence and arousal predicting decisions, negative valence carries all of the information, with a smooth increase in reject probability with increasing negativity – and little effect of arousal or specific emotion category. If so, this would challenge the conclusion that "negative valence and high arousal affective states are unlikely to motivate punishment", which I have to admit is implausible to me.

Direct associations with multiple emotion categories? One thing that seems to be missing from the presentation is a simple descriptive analysis of the proportion of trials on which people report each

emotion when they accept versus reject offers. The intervening machine learning models are right some of the time but wrong most of the time (e.g., 70% of the time) in picking the correct emotion label. So the empirical mapping between emotion category and accept/reject decision could be different (e.g., sparser in emotion space) than the “smoothing” done by the neural network implies. Put another way, the result that “the top three emotions associated with decisions to punish are sadness (13.47%), disappointment (12.82%), and disgust (12.54%)” is based on model likelihoods, but the ground truth of what people reported is available, and it could reveal a different pattern.

In a similar vein, it would be helpful to have inferential statistics for each emotion on the likelihood of acceptance/rejection conditional on the label. This kind of basic statistical test could provide more solid statistical links between emotion category labels and decisions, but it seems to be absent in this version.

Ambiguity caused by picking single emotion labels

Emotion classification may rely on requiring participants to choose a single emotion category label? If so then when people report they are feeling disgusted or disappointed could they also be feeling angry? This would also challenge the main conclusions.

Points of uncertainty

How good are people at using the affect grid in this online context? How much of the diversity in emotion reports is measurement noise?

I don't understand something about the clustering results. A finding is: “When deciding to accept the offer, cluster 5 (neutral valence and arousal; “neutral”) was almost twice as likely ($\chi^2(1)=520.22, p<.001$) as cluster 2 (neutral arousal and positive valence; “satisfied”)” There are very few figures showing any actual study data or inferential statistics, which is a limitation. But assuming Figure 1C shows the actual accept/reject data, this result is puzzling: The Cluster 2 area looks to be nearly 100% Accept, where as Cluster 5 is mixed, with reject on the negative valence side and accept on the positive valence side – matching intuition, but not the result above.

Does the arousal and valence rating set do better at predicting ultimatum game choices than the more complex set of category labels? Do category labels add anything in a fair test? (see above)

How granular are the accept/pun decisions? I'm assuming they are binary.

Minor comments

“high overall accuracy (NN: 35.8% and KNN: 36.0%, compared to null accuracy of 5%)”

I'm assuming this is in 20-way classification of emotion label. But what is the unit of analysis/aggregation here? i.e., A single trial or an average of a set of trials?

There seems to be strong agreement between the neural network and the KNN algorithm in parsing the valence/arousal space, which is good.

The fact that the k-means clustering partitioned balance and arousal space into evenly spaced cubes probably means that there are no distinct clusters.

Details

Author info missing from ref six

Journal info ref 15

interrogating how much anger a person feels in response to unfair treatment may artificially impose an expectation that the person ought to feel anger 30

Work by Feldman Barrett is relevant here

attributing specific emotional states to increased physiological and neural activity may falsely lead to mis-identifying the emotion actually experienced 31.

Work by Khan et al. 2002, Kober et. al. 2008, Lindquist et al. 2012 relates to this point specifically in the domain of emotion.

“riding at the expensive of the common good”

Reviewer #2 (Remarks to the Author):

This paper presents a set of three studies examining individuals' affective reactions to antisocial choices in competitive social interactions (here operationalized using ultimatum game, prisoner's dilemma, and public goods game tasks). Participants rated a set of emotion terms on valence and arousal by placing these in 2D affect grid; these data were then used to train and test classifiers that identify the regions of the affect grid associated with each emotion term. When applied to affective rating data gathered during the tasks, these classifiers indicated that regions associated with a number of negative emotions are associated with antisocial choices. Critically, the region associated with 'anger' ratings was rarely among the most common. These findings broadly held across all three tasks and were supported by unsupervised clustering analyses using only the in-task rating data.

My thanks to the authors for the opportunity to review this work. This a targeted, well-designed, and cohesive set of studies. The analytical approach is comprehensive and leverages modern methods without becoming overbearing or opaque. The narrative is tight and the paper is very well written and visualized. I have a few suggestions regarding clarification and interpretation.

1. It could be, as noted on lines 268-9, that “framing anger as a motivating force may mischaracterize the nature of the relationship” between emotion and choice. Yet it could also be that framing anger as a unitary phenomenon (i.e., essentializing it) is mischaracterizing the nature of the category. There is work by Kuppens and colleagues (2003, 2007), for example, that explores heterogeneity in features of anger. Harmon-Jones et al. (2009) show that instances of anger can be pleasant (see also work by Wilson-Mendenhall and colleagues exploring within-category variation in affect). Indeed, the authors make passing reference to the ‘types of anger’ (e.g., quiet anger) that participants could represent and rate in the emotion classification task. What implications do the current findings/approach have for our understanding of emotion categories in general?

2. Relatedly, there is an assumption being made that emotion categories can be sufficiently represented in 2D valence/arousal space. This assumption seems reasonable for present purposes, as the authors provide data to suggest that the supervised analyses are still able to recover meaningful class boundaries. Still, I think there could be a bit of discussion about what exactly is being discriminated, and how fully this corresponds to these categories/concepts. For example, disgust is also a negative, higher arousal emotion. It is not strictly possible to know whether a participant was representing disgust vs. a lower-arousal version of anger when they provided their rating in a given instance. How does this shape interpretation of the current findings, or how do the authors think future research can incorporate other features or dimensions of emotions?

3. Can the authors clarify why they had participants complete the emotion classification task before the game, and what the potential impacts of this decision might be? The same question holds for the decision to have them provide affect ratings before deciding to accept, reject, etc.

4. Smaller comments:

a. Line 160: Clarify how ‘unfairness’ was operationalized.

b. It wasn't completely clear to me how these Euclidean distance analyses worked before I read the supplemental material, so perhaps a bit more detail could added inline.

- c. I don't see the continuous analyses for Studies 2 and 3 mentioned in the main text.
- d. Can the authors provide more information about the criteria they used to cross-validate their supervised models?

Reviewer #3 (Remarks to the Author):

The authors conducted 3 studies using samples from Mechanical Turk in which participants made decisions as a recipient in the UBG, and sequentially after learning about their partners behaviors in a PD and PGG. The authors measured how people felt about their partners behavior using an affect grid, which doesn't rely on measuring emotions with emotion labels. The participants prior to the social decision-making task associated specific emotions labels with responses to the affect grid. The authors used various machine learning approaches to associate responses to the affect grid with behavioral responses to their partner's behavior, and to infer which emotions were associated with these behaviors. The authors found that high valence, negative emotions, such as anger, were much less likely to be associated with punishment and defection behaviors compared to more neutral negative emotions, such as disappointment and sadness. I think the authors use a novel method to test the idea that anger is the emotion underlying these behaviors in these experiments. The data certainly challenge that assumption. I offer several comments that I hope the authors find constructive in further improving the manuscript.

Is the affect grid method a validated measure for anger – such that people who actually experience anger – they can rate anger using this measure? The authors do ask people who are not feeling an emotion to place this emotion on the affect grid, but saying how they would rate that emotional experience may not be the same as actually feeling that emotion. This research could benefit from a validation study to establish construct validity for specific emotions, or to cite research that has already used the affect grid and established construct validity for the experience of anger (and other emotions).

The authors conclude in experiment 1 that "... they provide converging evidence that negative valence and high arousal affective states are unlikely to motivate punishment." But was cluster 1 a significant predictor of punishment in that experiment? And was Cluster 3 simply a stronger predictor of punishment? If so, then shouldn't the conclusion be that emotions other than anger, such as disappointment, are a stronger predictor of punishment behavior, compared to anger (not that anger does not predict punishment).

The authors find that "Results reveal anger as the 3rd most likely emotion to be experienced (7.97%), with disgust (9.41%) and disappointment (8.15%) found to be significantly more likely (paired t-tests, all $P_s < .001$)." These data are perhaps the strongest results presented in the paper, and the conclusion is that disgust and disappointment are also as important as anger. I don't see much differences between these estimates. Is disgust even statistically different than anger in this case?

There is actually a lot of work which has studied disappointment and responses to partner behaviors in these kinds of games. I actually was disappointed that the authors did not cite or reference this work. It is not such a novel claim or discovery that disappointment is experienced in these contexts. The method is novel and the findings still important though. And indeed, many people still do claim that anger is the emotion underlying these actions. But I do think the authors should acknowledge the literature on disappointment in response to others non-cooperative behaviors.

Wubben, M. J., De Cremer, D., & Van Dijk, E. (2009). How emotion communication guides reciprocity: Establishing cooperation through disappointment and anger. *Journal of experimental social psychology*, 45(4), 987-990.

Van Doorn, E. A., Heerdink, M. W., & Van Kleef, G. A. (2012). Emotion and the construal of social situations: Inferences of cooperation versus competition from expressions of anger, happiness, and disappointment. *Cognition & emotion*, 26(3), 442-461.

Wubben, M. J., De Cremer, D., & van Dijk, E. (2011). The communication of anger and

disappointment helps to establish cooperation through indirect reciprocity. *Journal of Economic Psychology*, 32(3), 489-501.

Van Kleef, G. A., & Van Lange, P. A. (2008). What other's disappointment may do to selfish people: Emotion and social value orientation in a negotiation context. *Personality and Social Psychology Bulletin*, 34(8), 1084-1095.

Martinez, L. F., & Zeelenberg, M. (2015). Trust me (or not): Regret and disappointment in experimental economic games. *Decision*, 2(2), 118.

The authors bin behavior in the PD in Experiment 2 into a dichotomous variable (e.g., 0-49 is defect). And this approach was also used in experiment 3 with the PGG. I would hardly consider a contribution of .40 as defecting in these games. In fact, that is about the average amount of contributions in an experiment like this. The authors report the results of continuous regression models in the SI and I would recommend that these are the main results reported in the manuscript.

Relatedly, in the PD game, participants made their decision in a sequential order. I think it would be important to study how emotions were associated with the differences in contributions. For example, do people who experience anger versus disappointment are more likely to give even less than their partner gave them. Also, what emotions are associated with conditional cooperation and matching a partner's behavior?

I recommend that the authors not label these behaviors as "antisocial". Defection can be a "good" behavior, if people defect against another person who has a bad reputation or who has behaved poorly in the past (as can be the case in these studies). I would recommend labeling these behaviors as punishment behaviors, or decisions to not cooperate (defect).

The authors study how people respond to many different outcomes and find that high valence emotions are not strongly associated with punishment (defection) decisions. I wonder whether this finding is an artifact of the experimental tasking being repetitive, boring, and just not very engaging. This could be a point raised in the discussion. To be clear, that is a critique that applies to this field of study, generally, and it's not something unique to the methods used here. But I do wonder if people had only made a single decision that determined their entire outcome of the experimental session, that this would elicit more high valence emotions, compared to making this same kind of decision repeatedly.

I am unable to comment on the different machine learning approaches to analyzing the data. However, I was curious whether 40% accuracy is really an acceptable standard in this area of work. These analyses seem to have a lot of error in classifying an emotion, but I am not expert on these methods and evaluating a "good" model.

Reviewer 1

Positive comments

This paper takes a fresh and innovative look at how self-reported emotion categories map onto social decision making, and in particular punishment of others during social interactions. The main thrust of the paper seems to challenge the link between anger and social punishment (rejecting unfair offers in the Ultimatum Game, defection in Prisoner's Dilemma and Public Goods games), by claiming that a broader set of emotions (e.g., nervous, afraid, or even sluggish) are also associated with punishment. This is used to argue that anger is less important as a motivator of punishment than researchers previously thought.

The concept is clear and the paper blends emotion research with machine learning models in an accessible way. The figures are informative and aesthetically beautiful. (But: There are very few figures showing any actual study data or inferential statistics, which is an important limitation for me).

The conclusions are surprising and would be interesting if borne out: e.g. "negative valence and high arousal affective states are unlikely to motivate punishment". But (unfortunately) I'm not convinced that they're warranted based on the present manuscript.

Potential weaknesses

Major concerns

Comment 1: *Theoretical claims. This paper seems to imply that because people punish when they're angry, the literature suggests that people reject offers only when they're angry. The paper challenges the latter claim, but not the former. I'm not sure that many emotion theorists would endorse the latter. For example, with many people really endorse the idea that people reject an unfair offer when they are angered by it but do not reject disappointed or saddened by it? Put another way, I can imagine many situations in which people are disappointed or disgusted by propositions and not truly "angered", but still reject those propositions. Some such propositions were studied here – e.g., being offered \$3 out of a possible \$10 may be "disappointing", and I may reject the proposal. $P(\text{reject} \mid \text{disappointment})$ may be high. But this does not inform on whether $P(\text{reject} \mid \text{anger})$ is high. It does inform on the reverse inference, $P(\text{anger} \mid \text{reject})$, i.e., that I reject only when angry. But I do not believe the literature makes any strong claim about this hypothesis, and high $P(\text{anger} \mid \text{reject})$ is not very plausible in the first place.*

Response 1: Thank you for highlighting this important theoretical point, which we did not sufficiently explain in our original submission. We did not mean to imply that the existing literature argues people reject an unfair offer *only* when angered by it, and therefore do not reject offers when feeling disappointment or sadness. To date, the majority of research in behavioral economics and psychology focuses on the relationship between anger and punishment, although there is some work that explores the relationship between punishment and other emotions (such as guilt, regret, and disappointment). That the prevailing narrative emphasizes anger, does not—as the reviewer points out—rule out the involvement of other emotions motivating decisions to punish. Our goal aligns with this comment: we took an agnostic, unbiased view about the relationship between specific emotions and punishment, and let a data-driven approach reveal the nature of this relationship. We have now clarified this point in the introduction (page 2-3) and also now cite prior literature that has examined the relationship between other emotions and punishment. In line with this, we completely agree that the conditional probability of reject—given a feeling of disappointment, does not necessarily inform the conditional probability of reject—given a feeling of anger. However, one of the strengths of our paradigm is that we can report the model likelihoods in a variety of ways, and we have added new visualizations of these likelihoods to the

manuscript (Figure 4, page 26). See response 5 to Reviewer 1 below for a more detailed discussion on this point.

Comment 2: *Alternative models. The paper maps rejection of offers (social punishment) onto a relatively complex space of emotions. The implication is that these emotions are important, even potentially as causal factors that impact decision-making. But what if the decision space is much simpler? For example, responders in the ultimatum game might choose to reject offers based on a policy decision, if they are below 20% of the total pie, for example. This is an example of a much simpler model. Does a policy model explain the data better than a model that requires knowing whether one is sad, disappointed, angry, or sluggish?*

Response 2: The Reviewer brings up an important point about alternative models of decision-making in the Ultimatum Game (UG), models which may not necessarily require (or be an improvement over) a model that includes emotional reactions. To address this fair critique, we used mixed-effects logistic regressions to test two models of decision-making in the UG: one which models choices to punish as a function of unfairness (1a), and one which models choices to punish as a function of unfairness interacting with affective experiences (valence and arousal separately). Model 1a captures the intuition that people use a policy decision (e.g., reject if below 20% of the total pie) as the slope of unfairness (β_1) will represent the strength of the policy decision while the intercept (β_0) will represent the point of indifference where participants are 50% likely to punish (vs accept). Model 1b captures the intuition that affective experiences improve predicting decisions to punish by adjusting one’s sensitivity to unfair offers. We note that because the probabilities add up to 1, we cannot include the probability of every emotion category in the same model (e.g., choice ~ sad + anger + etc.), which would create a fixed-effect model matrix that is rank deficient. However, because the emotion categories (e.g., sad, disappointed, angry) are inferred from these very affect ratings, model 1b addresses the concern while still being interpretable.

The fixed-effects equations are shown below:

$$punish \sim \beta_0 + \beta_1 unfairness \tag{1a}$$

$$punish \sim \beta_0 + \beta_1 valence + \beta_2 arousal + \beta_3 unfairness + \beta_4 valence * unfairness + \beta_5 arousal * unfairness + \beta_6 valence * arousal * unfairness \tag{1b}$$

We used likelihood ratio tests to compare models and infer which model better explains decisions to punish. Likelihood ratio tests reveal that the model which includes affective experiences (1b) significantly improves the fit compared to the model which only included unfairness (m1a; $\chi^2(11) = 489.21, p < .001$). Results from model 1b show significant marginal effects of unfairness and valence such that greater unfairness increases the probability of rejecting while more positive valence decreases the probability of rejecting (Table 1). In short, a simpler model based purely on a decision policy of amount of money offered does a worse job of explaining behavior compared to a model that includes affective experiences. This result is now shown in Supplementary Table 4 on pages 8-9 of the SI.

Ultimatum Game Affect Model of Punishment

Predictors	Estimates			
	Log-Odds	std. Error	Statistic	p
Intercept	-4.71	0.28	-16.61	<0.001

Unfairness	3.85	0.26	14.95	<0.001
Valence	-2.28	0.22	-10.31	<0.001
Arousal	-0.00	0.18	-0.02	0.982
Unfairness:Valence	-0.13	0.15	-0.91	0.363
Unfairness:Arousal	0.19	0.13	1.48	0.138
Valence:Arousal	-0.09	0.14	-0.62	0.535
Unfairness:Valence:Arousal	0.20	0.11	1.88	0.060
N _{sub}	715			

Table 1. Ultimatum Game Affect Model of Punishment. Valence and arousal have been scaled, but not mean-centered, as the 0 point refers to the meaningful instance when affect is neutral. Unfairness is operationalized as the amount of money kept by Player B and has been normalized (scaled and mean-centered). The model included a random intercept and random slopes for unfairness, valence, and arousal per subject.

Comment 3: Likewise, is position in valence/arousal space a stronger predictor of accept/reject decisions than the emotion category labels? Does the reported category add predictive information, or is it superfluous? I suspect that in a logistic (or nonparametric logistic) regression with valence and arousal predicting decisions, negative valence carries all of the information, with a smooth increase in reject probability with increasing negativity – and little effect of arousal or specific emotion category. If so, this would challenge the conclusion that “negative valence and high arousal affective states are unlikely to motivate punishment”, which I have to admit is implausible to me.

Response 3: The reviewer highlights two points. First, “*is position in valence/arousal space a stronger predictor of accept/reject decisions than the emotion category labels?*” While we can examine how valence and arousal combine to predict decisions to punish (using an approach explained in the response above), we cannot test whether these affective ratings do a better job of predicting decisions compared to emotion category labels. This is because we collected participants’ responses in the emotion classification task (where a participant places their cursor at a specific spot on the affect grid that corresponds to an [X, Y] coordinate to denote the valence/arousal experience of a given emotion), and a separate set of responses using the same affect grid during the UG. In other words, we never asked participants to self-report an emotion category label during the UG, primarily because of the issues that arise when asking for self-reported emotional experiences (discussed in the paper on pages 2-3, and now additionally on 10-11). Instead, we use the affective responses in the emotion classification task to build a classifier, which we then apply to the unlabeled affect ratings elicited during the UG. In this way, we can *infer* categorical emotion experiences without ever directly asking about specific emotion terms. In short, because the unlabeled affect ratings are the only emotion measures recorded from the UG, we cannot test how these unlabeled responses fair compared to self-reported responses that are labeled.

That being said, we can answer the reviewer’s second question: “*does negative valence carries all of the information, with a smooth increase in reject probability with increasing negativity – and little effect of arousal or specific emotion category?*” Using a mixed-effects regression, we tested whether the

interaction between valence and arousal predicts decisions to punish using the following fixed-effects formulation:

$$punish \sim \beta_0 + \beta_1 valence + \beta_2 arousal + \beta_3 valence * arousal \quad (2)$$

Results from this mixed-effects regression (Table 2, appended below) show that while arousal on its own does not predict decisions to punish, there is a significant interaction between valence and arousal in predicting decisions to punish. This suggests that negative valence does not carry all the information in predicting decisions, but rather the joint combination of valence and arousal are critical in understanding how emotions relate to punishment. Given the variability in arousal ratings associated with decisions to punish (see Figure 1C of the manuscript), we also tested the possibility that arousal might predict decisions to punish in a non-linear way. We added a quadratic term for arousal to model 2 and results show a significant non-linear effect of arousal, such that increases in the quadratic value of arousal strongly predicts punishment (Table 3 below). Figure 1 below visualizes the main effect of arousal in this model, illustrating how the probability of punishing increases at the two extremes of the arousal scale. Furthermore, there is a significant interaction between valence and arousal which is visualized in Figure 2 below. This interaction reveals how the relationship between valence and punishment is significantly attenuated when arousal is neutral (arousal=0) compared to when arousal is extremely low (arousal=-2) or extremely high (arousal=+2; Figure 2 below). We now include these results as Supplementary Tables 5 and 6 and Supplementary Figures 4 and 5 on pages 9-11. Further, based on this comment and the new analyses we have revised this section of the manuscript, as we agree with the reviewer that our language was misleading. We thank the Reviewer for this suggested analysis and believe that these results have strengthened the manuscript by demonstrating the importance of both valence and arousal dimensions when explaining decisions to punish.

Interaction between Valence and Arousal Predicts Punishment

Predictors	Estimates			
	Log-Odds	std. Error	Statistic	p
Intercept	-3.88	0.18	-21.64	<0.001
Valence	-4.51	0.18	-25.51	<0.001
Arousal	-0.23	0.19	-1.22	0.222
Valence:Arousal	-0.38	0.16	-2.42	0.015
N _{sub}	715			

Table 2. All variables were scaled but not mean-centered, as the 0 point on each scale refers to the meaningful instance where affect is neutral. The model includes subject-specific random intercepts and slopes for Valence, Arousal, and their interaction.

Non-linear Effects of Arousal Predicting Punishment

Predictors	Estimates			
	Log-Odds	std. Error	Statistic	p
Intercept	-3.43	0.16	-20.89	<0.001
Valence	-4.20	0.17	-24.16	<0.001

Arousal(1)	7.33	11.91	0.62	0.538
Arousal(2)	66.66	10.05	6.63	<0.001
Valence:Arousal(1)	-24.75	9.36	-2.64	0.008
Valence:Arousal(2)	-12.80	8.92	-1.44	0.151
N _{sub}	715			

Table 3. All variables were scaled but not mean-centered, as the 0 point on each scale refers to the meaningful instance where affect is neutral. Orthogonal polynomials were used to test for non-linear effects of arousal and the degree of the polynomial is presented in paratheses (e.g., (1), (2)). The model includes subject-specific random intercepts and slopes for valence and the orthogonal polynomial terms for Arousal.

Figure 1. Main effect of Arousal from Table 3. The predicted regression line for the main effect of arousal from Table 3 is plotted for continuous values of arousal when valence equals 0. Valence and arousal were scaled, but not mean-centered, as the 0 point meaningfully refers to the case when affect is neutral. Error bars are +/- 1 standard errors of the mean.

Figure 2. Interaction between Valence and Arousal from Table 3. Predicted regression lines from Table 3 are plotted for continuous values of valence and binned values of arousal. For visualization purposes arousal has been binned to be -2, 0, or +2 standard deviation, although the regression used the continuous value. Valence and arousal were scaled, but not mean-centered, as the 0 point meaningfully refers to the case when affect is neutral. Error bars are +/- 1 standard errors of the mean.

***Comment 4:** Direct associations with multiple emotion categories? One thing that seems to be missing from the presentation is a simple descriptive analysis of the proportion of trials on which people report each emotion when they accept versus reject offers. The intervening machine learning models are right some of the time but wrong most of the time (e.g., 70% of the time) in picking the correct emotion label. So the empirical mapping between emotion category and accept/reject decision could be different (e.g., sparser in emotion space) than the “smoothing” done by the neural network implies. Put another way, the result that “the top three emotions associated with decisions to punish are sadness (13.47%), disappointment (12.82%), and disgust (12.54%)” is based on model likelihoods, but the ground truth of what people reported is available, and it could reveal a different pattern*

Response 4: We apologize for not being clearer in our original manuscript. To clarify, participants never self-reported any categorical emotional experience (e.g., “I feel sad” or “I feel disappointed”) during the Ultimatum Game. Instead, participants were asked to make affective ratings (valence and arousal) free of any emotion labels, and we classified these label-free affect ratings into categorical emotions based on the data from the emotion classifier task and the trained neural network. Therefore, we cannot present a descriptive analysis of the proportion of trials where participants reported feeling a specific emotion during the UG. The primary reason we do not ask for self-reported emotions using labels during the UG is because the use of emotion labels is associated with demand characteristics, such that participants are limited by reporting the strength of the emotions assumed to be associated with the task at hand (e.g., reporting how much anger they feel because the question only probes anger, even if they don’t actually feel angry). We have now included more discussion about why we did not ask for self-report emotions, so this point is more clear in the manuscript (see pages 10-11).

Comment 5: In a similar vein, it would be helpful to have inferential statistics for each emotion on the likelihood of acceptance/rejection conditional on the label. This kind of basic statistical test could provide more solid statistical links between emotion category labels and decisions, but it seems to be absent in this version.

Response 5: We agree with the Reviewer that visualizing the likelihood of acceptance / rejection conditional on the emotion label is helpful. We have added numbers indicating the model likelihoods into Figure 4A in the manuscript (page 26), and have also created another visualization (Figure 4B in the manuscript) which illustrates the difference in model likelihood for punish minus accept, given a specific emotion. This measure represents the specificity with which a specific emotion label is more likely to be associated with punishing compared to accepting. For example, when considering decisions to punish, the average likelihood that person experienced anger is 10.28%, whereas only there is only a 2.21% likelihood that a person experienced anger when deciding to accept. Thus, the experience of anger favors punishment (over acceptance) by 8.07%. The higher the difference, the more an emotion is associated with a particular decision. The emotions with the greatest differences between decisions to punish and accept are sadness (difference = 9.92%), disgust (9.11%), anger (8.07%), and disappointment (7.99%), see Figure 4B in the manuscript. We also include the model likelihoods for punish and accept separately in Figure 3 below, which is now Supplementary Figure 6 on pages 12-13.

Figure 3. Conditional probabilities of choice by neural network emotion classifications. The neural network model trained on the emotion classification data was applied to the unlabeled affect ratings from the Ultimatum Game. Each data point was assigned a probability of each emotion class and these were averaged within participants, and then across choices. Emotions are organized by the conditional probability of choice given emotion for each emotion class. Errors bars reflect 95% CIs.

Comment 6: Ambiguity caused by picking single emotion labels. Emotion classification may rely on requiring participants to choose a single emotion category label? If so then when people report they are

feeling disgusted or disappointed could they also be feeling angry? This would also challenge the main conclusions.

Response 6: This comment made us realize that we did not do an adequate job of describing what we did in our task, and that there is some confusion about how the paradigm was set up—and for this we apologize. Participants never selected emotion categories in either the classification task or during the economic games. The only time participants were presented with labelled emotions was during the emotion classification task, where they were asked to place each emotion label (e.g., ‘anger’) on the affect grid according to their memories and knowledge associated with each word. In other words, for each subject, ‘anger’ is linked to a specific [X, Y] coordinate that corresponds to a [valence, arousal] rating. Therefore, participants were never required to choose emotion categories, but instead rated each emotion term in a 2-dimensional coordinate system representing valence and arousal dimensions. When we trained a neural network to classify which emotions were being experienced during the economic game, we treated the valence and arousal ratings as input features and the emotion labels as the output class for the neural network. The neural network learned the mapping between input and output in the emotion classification task by probabilistically computing the likelihood of any given emotion label (output) based on the associated coordinates (input) made by the participant. Once trained, we then applied this model to the affect ratings made during the economic games where we had the input features but not the output (emotion labels). Thus, if a participant rated their affect experience in the Ultimatum Game in the upper left hand corner—say, on the coordinates (-200 valence, +150 arousal)—then the neural network assesses the likelihood of that experience compared to where the participant placed anger in the emotion classification task (or disappointment or disgust, etc.). The probability maps for each emotion given the dimensions of valence and arousal can be found in Figure 2C in the manuscript on page 24. Below, in Figure 4, we show the model likelihoods associated with the particular affective experience of (-200, +150) for illustrative purposes, which reveals how these particular coordinates are computed into probabilities associated with specific emotions. We have clarified the design in the methods, and have revised Figure 1 (see page 23) to better describe what was done in the task.

Figure 4. Model Likelihoods of Each Emotion Term Associated with the exact coordinate (-200, +150). This example uses a valence rating of -200 and an arousal rating of +150, which would be in the upper left-hand corner of the affect grid. The trained neural network computes a probability or likelihood that this affect experience is associated with each of the 20 emotion classes.

Comment 7: Points of uncertainty. How good are people at using the affect grid in this online context? How much of the diversity in emotion reports is measurement noise?

Response 7: This is an interesting question. The affect grid is often used in laboratory settings and has been shown to have good reliability, convergent validity, and discriminant validity when compared to the traditional single-item measures (e.g., asking for valence and arousal on separate Likert scales; Russell, Weiss & Mendelsohn, 1989). Furthermore, single-item valence and arousal ratings are very popular and well-validated for online studies (e.g., Kurdi, Lozano, & Banaji; Cowen & Keltner, 2017). We selected 16 of the 20 emotion labels for the emotion classification task to fully represent the quadrants of the circumplex (Feldman, 1995), and we find the group-level responses are consistent with the locations of those terms in prior laboratory studies (Feldman, 1995; Bliss-Moreau, Williams, Santistevan, 2019).

Although measurement noise is always a concern, it is theoretically difficult to distinguish ‘noise’ from individual (i.e., subjective) affective responses. This is because each person’s experience of anger, for example, is unique and idiosyncratic. One person might place anger in the upper left-hand corner of the grid, which would reflect that their experience of anger is very negatively valenced and highly arousing—something akin to rage. Another subject might place anger lower in the left-hand quadrant (where arousal is close to 0), which would suggest that this person’s experience of anger is slightly less negative and arousing—perhaps something that better reflects ‘quite anger’. These individual differences cannot be chalked up to ‘noise’ per se, but rather to the fact that each person’s experience of emotion is different. That is, while it may be tempting to argue that this is an error or measurement noise, these responses represent the diversity in affect ratings of the same discrete emotion categories. We have added a new figure which highlights this conceptual point (see Figure 2 in the manuscript, also appended below) and also added 2D and 1D density plots from the emotion classification task to better visualize this diversity of emotional responses (see Response 1 to Reviewer 2 below).

That being said, one place where we can mitigate issues of noise is when the neural network is computing its probabilistic labels. By using a 10-fold cross-validation to train the neural network, we ensure that the classification labels, and therefore the model likelihoods, are as robust as possible to random noise associated with responding using the affect grid (e.g., one subject rating anger in the upper right-hand corner of the grid, something akin to ‘satisfaction’). To be more transparent about the confidence in the results not being driven by noise of the neural network, we have added effect sizes (Cohen’s d ’s) to all main results which may help readers better assess the meaningfulness of these differences.

Figure 5. Emotion classification Plots. A) Two-dimensional density plots of where participants placed each emotion term in the emotion classification task. The x-axis represents valence and the y-axis represents arousal. Contour lines illustrate different levels of density at the population level, see Supplement for more detailed visualizations. **B) One-dimensional density plots illustrating** the ratings of emotion terms in the emotion classification task, plotted separately for valence and arousal.

Comment 8: *I don't understand something about the clustering results. A finding is: "When deciding to accept the offer, cluster 5 (neutral valence and arousal; "neutral") was almost twice as likely ($\chi^2(1)=520.22, p < .001$) as cluster 2 (neutral arousal and positive valence; "satisfied")" There are very few figures showing any actual study data or inferential statistics, which is a limitation. But assuming Figure 1C shows the actual accept/reject data, this result is puzzling: The Cluster 2 area looks to be nearly 100% Accept, where as Cluster 5 is mixed, with reject on the negative valence side and accept on the positive valence side – matching intuition, but not the result above .*

Response 8: We thank the Reviewer for this point and agree that we ought to have included more visualizations of the study data and inferential statistics. We have now added two more figures of the results, which show the proportion of decisions to punish within each cluster (Figure 6A appended below) and the proportion of each decision (Accept/Punish) per cluster (Figure 6B appended below). The Reviewer is correct that the responses inside Cluster 2 make up nearly all of Accept decisions (97.4% Accept, 2.60% Punish) and likewise for Cluster 5 (84.4% Accept, 15.6% Punish). However, these are *relative* percentages and do not take into account what percentage of Accept or Punish decisions are made inside of Cluster 2 or Cluster 5 relative to *all* decisions (i.e., accounting for the base rate of punishing or accepting). When you look at all decisions to Accept, 20.9% of these choices are associated with affective responses that fall inside Cluster 2, whereas Cluster 5 accounts for 38.4% of all decisions to Accept. In

other words, more decisions to Accept end up in Cluster 5, even though Cluster 5 includes relatively more punishment decisions than those found in Cluster 2. We discuss this in the manuscript on pages 6-7 and include this figure as Supplementary Figure 7 on page 14 of the Supplement.

Figure 6. k-means clustering proportions. **A) Proportion of decisions to punish, sorted by cluster number.** Affective experiences in the Ultimatum Game were clustered into nine clusters using k-means clustering. Bars represent the proportion of punishment decisions within each cluster. Error bars reflect 95% CIs. **B) Proportion of clusters by choice.** Stacked bars represent the proportion of clusters which fall into decisions to accept or punish.

Comment 9: Does the arousal and valence rating set do better at predicting ultimatum game choices than the more complex set of category labels? Do category labels add anything in a fair test? (see above)

Response 9: We apologize for not being clearer about what participants did in the task, however this cannot be tested. Please see response 3 above for a details as to why this is the case.

Comment 10: How granular are the accept/pun decisions? I'm assuming they are binary.

Response 10: Accept and punish decisions are binary and we have clarified this in the methods of the manuscript. While we used binary choices in the Ultimatum Game, participants had continuous choices for the Prisoner's Dilemma and Public Goods Game.

Comment 11: "high overall accuracy (NN: 35.8% and KNN: 36.0%, compared to null accuracy of 5%)". I'm assuming this is in 20-way classification of emotion label. But what is the unit of analysis/aggregation here? i.e., A single trial or an average of a set of trials?

Response 11: We have clarified that this was a 20-way classification problem (null accuracy would be 5% which represents simply guessing (1/20 emotions) for each data point) on page 4 of the manuscript, and that the unit of analysis refers to the proportion of cases the neural network correctly labeled an emotion in the held out 25% testing dataset (e.g., 35.8%). In other words, the percentage (e.g., 35.8%) can be thought of as the average correct classification from the testing dataset.

Comment 12: There seems to be strong agreement between the neural network and the KNN algorithm in parsing the valence/arousal space, which is good.

Response 12: We agree this is a good sign, and one of the reasons we included all the models we tested.

Comment 13: The fact that the k-means clustering partitioned balance and arousal space into evenly spaced cubes probably means that there are no distinct clusters.

Response 13: While this is a fair comment, we remain agnostic to whether the clusters pulled out by the k-means clustering present “ground truth” clusters in emotions—as we believe that would be impossible to theoretically defend. The k-means clustering works by simply defining centroids of each cluster and minimizing intra-cluster variance. In other words, it adjusts the centroids of the clusters until converging on the minimum sum of squared distance between the data points and the center of their cluster. Although we did specify nine distinct clusters, we had no way of specifying how uniform each of the clusters would be. It is not particularly surprising that the k-means algorithm pulled out nine roughly evenly spaced clusters as the labels in the emotion classification task were specifically selected to evenly represent the affect space—which speaks to its validity as a method for measuring emotions.

Comment 14: Author info missing from ref six

Response 14: Thank you for the attention to detail, we have corrected this.

Comment 15: Journal info ref 15

Response 15: Thank you again for the close reading, we have fixed this error.

Comment 16: “interrogating how much anger a person feels in response to unfair treatment may artificially impose an expectation that the person ought to feel anger” Work by Feldman Barrett is relevant here. “attributing specific emotional states to increased physiological and neural activity may falsely lead to mis-identifying the emotion actually experienced” Work by Khan et al. 2002, Kober et. al. 2008, Lindquist et al. 2012 relates to this point specifically in the domain of emotion.

Response 16: Thank you for highlighting these references, we have added them to the manuscript in the appropriate places.

Comment 17: “riding at the expensive of the common good”

Response 17: We have corrected this sentence to read: “free-riding at the expense of the common good”. Thank you.

Reviewer 2

This paper presents a set of three studies examining individuals' affective reactions to antisocial choices in competitive social interactions (here operationalized using ultimatum game, prisoner's dilemma, and public goods game tasks). Participants rated a set of emotion terms on valence and arousal by placing these in 2D affect grid; these data were then used to train and test classifiers that identify the regions of the affect grid associated with each emotion term. When applied to affective rating data gathered during the tasks, these classifiers indicated that regions associated with a number of negative emotions are associated with antisocial choices. Critically, the region associated with 'anger' ratings was rarely among the most common. These findings broadly held across all three tasks and were supported by unsupervised clustering analyses using only the in-task rating data.

My thanks to the authors for the opportunity to review this work. This a targeted, well-designed, and cohesive set of studies. The analytical approach is comprehensive and leverages modern methods without becoming overbearing or opaque. The narrative is tight and the paper is very well written and visualized. I have a few suggestions regarding clarification and interpretation.

Major concerns

Comment 1: *It could be, as noted on lines 268-9, that "framing anger as a motivating force may mischaracterize the nature of the relationship" between emotion and choice. Yet it could also be that framing anger as a unitary phenomenon (i.e., essentializing it) is mischaracterizing the nature of the category. There is work by Kuppens and colleagues (2003, 2007), for example, that explores heterogeneity in features of anger. Harmon-Jones et al. (2009) show that instances of anger can be pleasant (see also work by Wilson-Mendenhall and colleagues exploring within-category variation in affect). Indeed, the authors make passing reference to the 'types of anger' (e.g., quiet anger) that participants could represent and rate in the emotion classification task. What implications do the current findings/approach have for our understanding of emotion categories in general?*

Response 1: We thank the Reviewer for their thoughtful question. There are two main implications that the current findings and approach have for understanding emotion categories in general. First, we believe this approach offers a valuable tool for emotion researchers who want to tackle emotion concepts in a way which avoids the demand characteristics associated with self-report. By using a data-driven, agnostic approach to classify affective reactions into categorically labeled emotions, researchers can test claims made that arise from different theories of emotions. As we now state in the discussion on pages 11-12, this flexible approach could even be expanded to include more than two dimensions, which could greatly improve the confidence in the classifications.

Second, as the reviewer points out, this approach is also ideally suited for illustrating the variability within emotion categories themselves. For example, how heterogenous are specific emotional experiences, such as anger, when compared to disappointment? We can address this question by visualizing the spread and variance of each emotion category using 2D (valence x arousal; Figure 5A appended above in Response 7 to Reviewer 1, and which we now have added to the manuscript as Figure 3A) and 1D (valence and arousal separately, Figure 5B above, which we added to the manuscript as Figure 3B) density plots. The 2D graphs show a unique view of the emotion classification data and corroborate the literature on anger that the Reviewer references: anger is densely centered in the high arousal, unpleasant affect space ('rage') but also has a small cluster of ratings which are neutral arousal but highly unpleasant ('quiet anger'). This suggests a qualitative difference in the experience some participants have when considering the emotion 'anger'. Other emotions, such as disappointment, have a much wider heterogeneity across the grid. It is likely that the highest density regions reflect the canonical emotion response associated with each category, but also reveal nuances which may be meaningful to specific emotion categories.

These differences can be meaningfully quantified depending on the expected distribution for an emotion category (Table 4 below), and visualized in the 1D density plots (separately plotted for valence and arousal) also appended above. For example, if an emotion (e.g., “quiet”) is thought to be relatively homogenous in the valence dimension, a researcher might expect a unimodal distribution and can quantify the variance in the valence ratings of “quiet” according to the standard deviation (74.65) or interquartile range (61.5). There are other situations however, when an emotion term may evoke different interpretations, in which case the distribution may have two (or more) peaked distributions of responses. If we take “relaxed” as an example of this, we observe a bimodal distribution. To test whether this multimodal distribution is significantly different from a unimodal distribution (as in the case of “quiet”), we can leverage a number of measures that are specifically used to distinguish between unimodal and bimodal responses. One particularly robust metric is known as the Hartigan’s dip statistic (HDS; Hartigan & Hartigan, 1985; Freeman & Dale, 2013), which is calculated by taking the maximum difference between the observed distribution and a unimodal distribution (often a uniform) that has the smallest value deviations (i.e., minimizing this maximum difference). A sampling distribution of these differences (known as a dip) is created using repeated sampling of this distribution (with the sample size of the original data). If the p-value of this “dip” statistic is at, or greater than 95th percentile among all sampled values, then the distribution is considered to be multimodal. In the case of “relaxed”, the distribution of the valence ratings is significantly different from a unimodal distribution ($D = 0.0550, p < .001$). If we assume that the distribution is bimodal, then one possible explanation for the observed distribution is that at the population level, different participants are reflecting upon distinct experiences when they consider the word “relaxed.” In other words, some participants rated this emotion as being neutral on valence (neither pleasant nor unpleasant), while others rated this experience as highly pleasant. We present these analyses in the Supplement (Supplementary Tables 1 and 2 on pages 5-6) only to demonstrate how this technique can be used to better understand the heterogeneity of these emotion categories. We have also added a paragraph to the results section that illustrates the heterogeneity of emotion categories (page 5 of the manuscript) and have added additional language into the discussion that sharpens these points and discusses the implications for the field of emotion. Simply put, we think this approach offers a powerful view of the affect space, and reveals how heterogenous (or not) a given emotion is at the population level, which can help to better quantify and test predictions that stem from emotion theory.

Emotion Label	Valence Distribution Type	Valence HDS
Afraid	Multimodal	0.015*
Angry	Unimodal	0.01
Annoyed	Multimodal	0.014*
Aroused	Multimodal	0.052***
Calm	Multimodal	0.026***
Disappointed	Unimodal	0.008
Disgusted	Unimodal	0.009
Enthusiastic	Multimodal	0.015*
Happy	Unimodal	0.012
Nervous	Multimodal	0.03***
Peppy	Multimodal	0.026***
Quiet	Multimodal	0.056***
Relaxed	Multimodal	0.055***
Sad	Unimodal	0.01
Satisfied	Unimodal	0.012
Sleepy	Multimodal	0.054***
Sluggish	Multimodal	0.02***
Still	Multimodal	0.07***
Surprised	Multimodal	0.042***

Table 4. Multimodality in valence experiences by emotion categories. Evidence for a multimodal distribution was determined through Hartigan’s dip statistic (HDS). HDS is a null-hypothesis test so a significant result indicates the rejection of a unimodal distribution. The distribution column indicates whether the valence distribution is unimodal or multimodal according to the HDS. All p-values are uncorrected for multiple comparisons.

Emotion Label	Arousal Distribution Type	Arousal HDS
Afraid	Multimodal	0.028***
Angry	Multimodal	0.039***
Annoyed	Multimodal	0.018**
Aroused	Unimodal	0.013
Calm	Multimodal	0.061***
Disappointed	Multimodal	0.015*
Disgusted	Multimodal	0.015*
Enthusiastic	Multimodal	0.025***
Happy	Multimodal	0.036***
Nervous	Multimodal	0.024***
Neutral	Multimodal	0.098***
Peppy	Multimodal	0.018**
Quiet	Multimodal	0.066***
Relaxed	Multimodal	0.065***
Sad	Multimodal	0.016**
Satisfied	Multimodal	0.024***
Sleepy	Multimodal	0.022***
Sluggish	Multimodal	0.016**
Still	Multimodal	0.067***
Surprised	Multimodal	0.02***

Table 5. Multimodality in arousal experiences by emotion categories. Evidence for a multimodal distribution was determined through Hartigan’s dip statistic (HDS). HDS is a null-hypothesis test so a significant result indicates the rejection of a unimodal distribution. The distribution column indicates whether the arousal distribution is unimodal or multimodal according to the HDS. All p-values are uncorrected for multiple comparisons.

Comment 2: Relatedly, there is an assumption being made that emotion categories can be sufficiently represented in 2D valence/arousal space. This assumption seems reasonable for present purposes, as the authors provide data to suggest that the supervised analyses are still able to recover meaningful class boundaries. Still, I think there could be a bit of discussion about what exactly is being discriminated, and how fully this corresponds to these categories/concepts. For example, disgust is also a negative, higher arousal emotion. It is not strictly possible to know whether a participant was representing disgust vs. a lower-arousal version of anger when they provided their rating in a given instance. How does this shape interpretation of the current findings, or how do the authors think future research can incorporate other features or dimensions of emotions?

Response 2: We greatly appreciate the Reviewer’s comment and acknowledge that the 2D valence-arousal space is only one potential space of emotions to explore. As the Reviewer suggests, disgust and anger are very different emotions conceptually, but they could be treated similarly in the low-dimensional affect space as they are both associated with high arousal, unpleasant valence (we now discuss this on pages 11-12 of the manuscript). We believe that future research can (and should) incorporate other

features or dimensions of emotions, which would impact this flexible data driven approach in two ways: adding more dimensions could 1) improve the classification accuracy of the neural network, giving greater confidence in the reverse engineering process; 2) enable the approach to more robustly differentiate emotions which are similar along one dimension, but are perhaps different along another dimension (e.g., anger and disgust, as the reviewer notes). One excellent starting place would be to take the dimensions laid out in a recent publication (Cowen & Keltner 2017), where participants were asked to rate emotionally evocative videos on 14 cognitive appraisal dimensions including valence, arousal, approach (“to what extent does this make you feel like this is something you would want to approach?”), effort (“to what extent does this make you feel like viewing this demands effort?”), safety (“to what extent does this make you feel a sense of safety?”), etc. It would be very likely that anger and disgust would differ on the approach dimension, with anger being an approach-motivated emotion and disgust being an avoidance-motivated emotion. While the 2D valence-arousal space offers a quick measurement of affect, which may be useful for fast decision-making paradigms, adding in more dimension ratings could give empirical legs to a problem that has been mostly discussed on theoretical terms. We have added in a discussion of these issues on pages 11-12.

Comment 3: Can the authors clarify why they had participants complete the emotion classification task before the game, and what the potential impacts of this decision might be? The same question holds for the decision to have them provide affect ratings before deciding to accept, reject, etc.

Response 3: Thank you for the opportunity to clarify the logic of our design. Participants completed the emotion classification task first because we wanted to ensure that the affect ratings in the Ultimatum Game were interpretable. We reasoned that participants would better understand how to use the 2D affect space after having thought and explored that space in the more intuitive emotion classification task. As far as we know, prior research has only used the affect grid for a single task and shown high validity to other measurement techniques including single-item Likert ratings of valence and arousal separately (Russell, Weiss, & Mendelsohn, 1989). Although unlikely, it is possible that when completing the economic game (which came second), participants potentially reflected upon a discrete, categorical emotion (e.g., “I feel disappointed”) and then tried to remember where they rated “disappointment” in the emotion classification task. While we acknowledge this potential limitation, even this interpretation would not change the main findings.

As to the decision to have subjects provide affect ratings before deciding to accept/reject, we chose to have participants rate their affective response prior to a decision because the affect rating is directly in response to the offer itself (“how do you feel about the offer”) and not to how they feel after making a choice. That is, we did not want to muddy the emotional space by having the subject’s emotional experience potentially change once they decided to reject or accept the offer (which likely changes the emotional experience since the participant has the ability to change the outcome). We have now clarified this point in the methods of the manuscript.

Comment 4: Line 160: Clarify how ‘unfairness’ was operationalized.

Response 4: Unfairness was operationalized numerically according to the amount of money kept by Player A, the proposer (now clarified on page 13 and in Figure 6 of the manuscript). In our design, we elicited the full range of offers from the proposer, ranging from the proposer keeping \$.50 (fair) to \$.95 (highly unfair) out of \$1, in \$.05 increments. This resulted in 10 levels of unfairness. We have updated the graphs in the manuscript displaying unfairness to include these more granular levels

Comment 5: It wasn’t completely clear to me how these Euclidean distance analyses worked before I read the supplemental material, so perhaps a bit more detail could added inline.

Response 5: Thank you for this helpful suggestion. We have included more information from the Supplement in the manuscript to clarify how the Euclidean distance analysis works. We now write, on page 16 “Although the Euclidean distance measurement used in the individual difference analyses are not formal machine learning models, they represent another way to classify emotions that are unique to each participant. We calculate the Euclidean distance between the 20 emotion terms in the emotion classification task and the unlabeled affective reports during each trial of the economic games. We convert Euclidean distance into a numeric probability using inverse distance weighting, where the probability is the inverse of the Euclidean distance for a given emotion over the sum of inverse Euclidean distances of all emotions. Smaller Euclidean distances indicate a higher probability that the unlabeled affective experience is similar to the valence and arousal rating of the emotion term (e.g., anger, disgust, etc.) for each participant separately.”.

Comment 6: I don't see the continuous analyses for Studies 2 and 3 mentioned in the main text.

Response 6: Thank you for pointing this out, we apologize for missing this and have now included in the manuscript where those results can be found in the supplement (pages 18-20 for Experiment 2 and 24-25 for Experiment 3).

Comment 7: Can the authors provide more information about the criteria they used to cross-validate their supervised models?

Response 7: Absolutely. We split the original dataset into a 70-30 train-test split. Within the training data, we use 10-fold cross-validation which randomly splits the training data set into 10 additional-subsets. One subset is reserved for validation while the model is trained on the other subsets according to the model's specific algorithm. The model is tested on the reserved subset and an accuracy score is recorded for this subset of data. In our classification problem, accuracy is simply the proportion of correct classifications over the total classifications. For example, a null accuracy would be 5% because this represents a model which simply guesses at each classification. Because our dataset is balanced (i.e., each of the emotion classes are equally represented), accuracy is a good metric for evaluating the model. This process is repeated until each of the 10 subsets have served as the validation set. Finally, we compute the average of the 10 recorded accuracies which is the cross-validation accuracy and serves as the final performance metric for the model. We have now included more details about the cross-validation procedure in the Methods section of the manuscript, on page 14.

Reviewer 3

The authors conducted 3 studies using samples from Mechanical Turk in which participants made decisions as a recipient in the UBG, and sequentially after learning about their partners behaviors in a PD and PGG. The authors measured how people felt about their partners behavior using an affect grid, which doesn't rely on measuring emotions with emotion labels. The participants prior to the social decision-making task associated specific emotions labels with responses to the affect grid. The authors used various machine learning approaches to associate responses to the affect grid with behavioral responses to their partner's behavior, and to infer which emotions were associated with these behaviors. The authors found that high valance, negative emotions, such as anger, were much less likely to be associated with punishment and defection behaviors compared to more neutral negative emotions, such as disappointment and sadness. I think the authors use a novel method to test the idea that anger is the emotion underlying these behaviors in these experiments. The data certainly challenge that assumption. I offer several comments that I hope the authors find constructive in further improving the manuscript.

Major concerns

Comment 1: *Is the affect grid method a validated measure for anger – such that people who actually experience anger – they can rate anger using this measure? The authors do ask people who are not feeling an emotion to place this emotion on the affect grid, but saying how they would rate that emotional experience may not be the same as actually feeling that emotion. This research could benefit from a validation study to establish construct validity for specific emotions, or to cite research that has already used the affect grid and established construct validity for the experience of anger (and other emotions).*

Response 1: We thank the Reviewer for this important question. The affect grid as a method for measuring emotion has high reliability, convergent validity, and discriminant validity, especially when compared to the traditional single-item measures (e.g., asking for valence and arousal on separate Likert scales). In one validation study (Russell, Weiss & Mendelsohn, 1989), the affect grid showed high pairwise correlations between three other emotion measurement techniques (all r 's > .91). For example, participants were asked to judge the similarity between emotion words (e.g., how similar are disgust and anger on a 1-7 Likert scale). Using these similarity responses, multidimensional scaling was then used to extract the underlying dimensions between the emotion words. This approach showed success in describing 1) current mood (Russell & Gobet 2012)—which hopefully helps attenuate the Reviewer's concern, 2) the meaning of emotion-related words (Russell, Weiss, & Mendelsohn, 1989), and 3) the feelings conveyed by facial expressions (Tseng et al., 2014). We have added these citations in the methods discussing the affect grid and thank the Reviewer for the thoughtful comment.

Comment 2: *The authors conclude in experiment 1 that "... they provide converging evidence that negative valence and high arousal affective states are unlikely to motivate punishment." But was cluster 1 a significant predictor of punishment in that experiment? And was Cluster 3 simply a stronger predictor of punishment? If so, then shouldn't the conclusion be that emotions other than anger, such as disappointment, are a stronger predictor of punishment behavior, compared to anger (not that anger does not predict punishment).*

Response 2: Yes! We thank the Reviewer for this astute point and agree with their assessment. We have updated our conclusion of those results to read on page 7 "...provide converging evidence that negative valence and high arousal affective states are less likely to motivate punishment"

Comment 3: *The authors find that "Results reveal anger as the 3rd most likely emotion to be experienced (7.97%), with disgust (9.41%) and disappointment (8.15%) found to be significantly more likely (paired t-tests, all P s < .001)." These data are perhaps the strongest results presented in the paper, and the conclusion is that disgust and disappointment are also as important as anger. I don't see much differences between these estimates. Is disgust even statistically different than anger in this case?*

Response 3: While these differences are small, some are statistically different. In the Euclidean distance analysis disgust is significantly more likely than anger ($t(558) = 3.50, p < .001; d = 0.15$), although disappointment is not significantly greater than anger ($t(558) = 0.42, p = .68; d = 0.02$). We have added effect sizes for all reported analyses to help readers understand the effect size differences between model likelihoods. Although this analysis is just one of the many different approaches we leveraged to connect the classifier data to the affective experiences in the economic games, the results when taken together reveal that anger is not the only (nor the most likely) emotion to be associated with punishment. We have revised the language in the manuscript so that this is more clearly stated.

Comment 4: *There is actually a lot of work which has studied disappointment and responses to partner behaviors in these kinds of games. I actually was disappointed that the authors did not cite or reference this work. It is not such a novel claim or discovery that disappointment is experienced in these contexts. The method is novel and the findings still important though. And indeed, many people still do claim that anger is the emotion underlying these actions. But I do think the authors should acknowledge the*

literature on disappointment in response to others non-cooperative behaviors.

Wubben, M. J., De Cremer, D., & Van Dijk, E. (2009). How emotion communication guides reciprocity: Establishing cooperation through disappointment and anger. *Journal of experimental social psychology*, 45(4), 987-990.

Van Doorn, E. A., Heerdink, M. W., & Van Kleef, G. A. (2012). Emotion and the construal of social situations: Inferences of cooperation versus competition from expressions of anger, happiness, and disappointment. *Cognition & emotion*, 26(3), 442-461.

Wubben, M. J., De Cremer, D., & van Dijk, E. (2011). The communication of anger and disappointment helps to establish cooperation through indirect reciprocity. *Journal of Economic Psychology*, 32(3), 489-501.

Van Kleef, G. A., & Van Lange, P. A. (2008). What other's disappointment may do to selfish people: Emotion and social value orientation in a negotiation context. *Personality and Social Psychology Bulletin*, 34(8), 1084-1095.

Martinez, L. F., & Zeelenberg, M. (2015). Trust me (or not): Regret and disappointment in experimental economic games. *Decision*, 2(2), 118.

Response 4: We greatly appreciate this comment and the Reviewer's suggested references on disappointment—which we admittedly failed to include in the original submission. We agree that these should be cited in the manuscript, and have now revised our introduction and discussion on emotions and decision-making to include these references and provide a more in-depth account that better describes the existing research.

Comment 5: *The authors bin behavior in the PD in Experiment 2 into a dichotomous variable (e.g., 0-49 is defect). And this approach was also used in experiment 3 with the PGG. I would hardly consider a contribution of .40 as defecting in these games. In fact, that is about the average amount of contributions in an experiment like this. The authors report the results of continuous regression models in the SI and I would recommend that these are the main results reported in the manuscript.*

Response 5: We appreciate the recommendation and considered it during the revision process. However, after trying to incorporate the continuous analysis into the main paper and finding that it disrupts the narrative flow, we respectfully request that this analysis be stated in the manuscript but believe that it should not serve as the primary analysis for two reasons. First, the dichotomous analysis is used in the other experiment, which provides continuity across all three experiments. Second, the continuous analysis requires a rather detailed level of description that makes the paper overly burdensome to read, given the simplicity of the main finding and the fact that it replicates when the analysis is dichotomized. We have clarified in our results that these analyses replicate when analyzed in a continuous fashion and refer interested readers to the Supplement (pages 18-20 and 24-25). We hope that the reviewer finds this satisfactory.

Comment 6: *Relatedly, in the PD game, participants made their decision in a sequential order. I think it would be important to study how emotions were associated with the differences in contributions. For example, do people who experience anger versus disappointment are more likely to give even less than their partner gave them. Also, what emotions are associated with conditional cooperation and matching a partner's behavior?*

Response 6: This is such a great question. We have now run these analyses for Experiments 2 and 3 (this cannot be assessed in Experiment 1 as it uses a binary decision to accept/reject in the UG). Because the relationship between conditional cooperation and neural network emotion probabilities is largely non-linear, we used separate LOESS regressions for each emotion term for the PD (Figure 7) and PGG (Figure 8). Results show that emotions have different relationships with conditional cooperation; for example, participants are very likely to feel happy when completely defecting against their partners (conditional

contribution = -1) while emotions such as angry are most likely when conditional cooperation is high (conditional contribution = +1). Although participants most often matched their partner's contribution, one explanation for these puzzling results is that some participants consistently contributed nothing (\$0) or everything (\$1), perhaps reflecting that they decided before seeing their partner's contribution. Because each non-linear relationship is different, it is difficult to make broad interpretations about emotion (in general) and its relationship with conditional cooperation. Instead, we include the data and graphs for those who may be interested in how a specific emotion relates to conditional cooperation (Supplementary Figure 13 on page 21 and Supplementary Figure 18 on page 26).

Figure 7. NN Probabilities by Conditional Cooperation in the PD. Conditional cooperation is the difference between the participant's contribution and their partner, broken into five bins to better visualize the non-linear relationship between conditional cooperation and neural network probabilities. 1 indicates that the participant contributed one dollar more than their partner, 0 indicates that the participant and partner contributed the same amount, -1 indicates that the participant contributed one dollar less than their partner. Red lines are loess regression lines using the continuous conditional cooperation measure for each emotion category.

Figure 8. NN Probabilities by Conditional Cooperation in the PGG. Conditional cooperation is the difference between the participant’s contribution and the average partner’s contribution (total contribution / 3). Conditional cooperation has been binned into five bars to visualize the non-linear relationship between conditional cooperation and neural network probabilities. 1 indicates that the participant contributed one dollar more than the average partner’s contribution, 0 indicates that the participant and the average partner contributed the same amount, -1 indicates that the participant contributed one dollar less than the average partner. Red lines are loess regression lines using the continuous conditional cooperation measure for each emotion category.

Comment 5: I recommend that the authors not label these behaviors as “antisocial”. Defection can be a “good” behavior, if people defect against another person who has a bad reputation or who has behaved poorly in the past (as can be the case in these studies). I would recommend labeling these behaviors as punishment behaviors, or decisions to not cooperate (defect).

Response 5: We agree with the Reviewer and have removed all use of the word “antisocial” and instead use punitive and uncooperative, as suggested, and when referring to all three types of decisions, we use the less value laden term ‘competitive social decision-making’.

Comment 6: The authors study how people respond to many different outcomes and find that high valence emotions are not strongly associated with punishment (defection) decisions. I wonder whether

this finding is an artifact of the experimental tasking being repetitive, boring, and just not very engaging. This could be a point raised in the discussion. To be clear, that is a critique that applies to this field of study, generally, and it's not something unique to the methods used here. But I do wonder if people had only made a single decision that determined their entire outcome of the experimental session, that this would elicit more high valence emotions, compared to making this same kind of decision repeatedly.

Response 6: This is a fair point. We tried to diminish these problems by limiting our design to a relatively small number of trial (e.g., 20 trials for the Ultimatum Game) and engage participants by presenting them with photographs of their partner's face. However, as the Reviewer points out, it is a valid critique of many behavioral economics paradigms and we address it by examining whether trial effects could explain our results. We first examined the possibility that either affect rating (valence or arousal), may have changed over time in predicting choices to punish. For example, if the task was boring or not engaging, then there should be a change in the affect experienced and punishment displayed at the beginning of the experiment compared to the end. We tested this possibility by running a mixed-effects regression examining the three-way interaction between valence, arousal, and trial number (20 trials in total). Results from this analysis show a marginally significant effect of trial number such that punishment increased over time (Table 6). In this case, the probability of punishing, controlling for affect such that valence and arousal are both 0, rises by less than a percent from the beginning to the end of the experiment (increases by 0.85%). Although there was not a significant interaction between arousal and trial number, there was for valence such that later trials increased the probability of punishing only when there was more negative valence (Figure 9, red line). We visualize this effect to demonstrate how small of an impact it has on punishment. When valence was very unpleasant (fixed at -2 SD), there was a 98.8% probability of punishing at the beginning of the trial and this rises to 99.8% probability of punishing by the end of the experiment (1% increase). Together, although there is evidence that trial number has some influence on punitive decisions, it would be hard to leverage this marginal effect to explain the full set of results presented in the manuscript.

We additionally took a closer look at the result that shows no interaction between valence and arousal over trials on decisions to punish (Table 6; $p=0.151$). To unpack this result, we examined the k-means clustering results in the beginning and end of the experiment. If the UG task is repetitive, boring, and not very engaging, we should see that negative valence and high arousal experiences are the predominant response associated with punishment at the beginning of the experiment when the task is still novel. Results show that cluster 1 (high arousal, negative valence) represents 21.2% of punishment decisions at the beginning of the experiment, while cluster 3 (medium arousal, negative valence) represented 40.5% (Figure 10). At the end of the experiment, these frequencies are slightly reduced to 18.5% for cluster 1 and 36.1% for cluster 3. Thus, assuming that the decisions at the beginning of the task are more engaging and meaningful than those later on, there does not seem to be evidence that participants become bored and disengaged by the end of the experiment.

Trial Effects in the UG

Predictors	Estimates			
	Log-Odds	std. Error	Statistic	p
Intercept	-3.97	0.19	-21.22	<0.001
Valence	-4.65	0.19	-24.01	<0.001
Arousal	-0.08	0.16	-0.49	0.622

Trial	0.14	0.08	1.71	0.086
Valence:Arousal	-0.25	0.11	-2.21	0.027
Valence:Trial	-0.22	0.09	-2.58	0.010
Arousal:Trial	-0.10	0.08	-1.33	0.185
Valence:Arousal:Trial	-0.09	0.07	-1.43	0.151
N _{sub}	715			

Table 6. Trial Effects in the Ultimatum Game. Participants completed 20 trials of the Ultimatum Game and trial was standardized (scaled and mean-centered) for this regression. Valence and arousal were scaled, but not mean-centered, as the 0 point meaningfully refers to the case when affect is neutral. The model included a random intercept and random slopes for valence, arousal, and trial.

Figure 9. Relationship between Punishment and Valence over Time. Participants completed 20 trials of the Ultimatum Game and trial was standardized (scaled and mean-centered). Predicted regression line comes from Table 5 where valence values were set to -2, 0, and +2 standard deviations and arousal values were set to 0 (neutral).

Figure 10. Clustering Results Separated by Beginning and End of Ultimatum Game. Clustering results were derived from the k-means clustering analysis. Results were binned into the beginning of the Ultimatum Game (first 5 trials) and the end (last 5 trials).

Comment 7: I am unable to comment on the different machine learning approaches to analyzing the data. However, I was curious whether 40% accuracy is really an acceptable standard in this area of work. These analyses seem to have a lot of error in classifying an emotion, but I am not expert on these methods and evaluating a “good” model.

Response 7: Although 40% may not appear on the face of it to be “highly” accurate, it is important to note that this is a difficult classification problem, where we are trying to achieve a 20-way classification with only two dimensions (valence and arousal). These constraints result in inherent difficulties for achieving high levels of accuracy—although what constitutes a “good” model is relative to the problem at hand. The null accuracy of the model is 5%, and thus reaching almost 40% accuracy when classifying one of twenty different emotions is pretty good. However, we acknowledge that these are subjective perspectives and have added in some discussion about how future research can improve classification accuracy by introducing additional dimensions to learn on. Please see Response 2 to Reviewer 2 and this discussion on page 12 of the manuscript for more details about these future directions.

REVIEWER COMMENTS

Reviewer #2 (Remarks to the Author):

I thank the authors for providing exceptionally thorough and well-considered responses to my previous comments (as well as to those of the other reviewers). I believe this paper is a strong contribution to the literature and provides conceptual and methodological tools that will drive further innovation in emotion and other research domains.

Reviewer #3 (Remarks to the Author):

I appreciate the authors' responses to my comments. I have no further comments. This manuscript is a fine addition to the literature. Congrats!

Response to Reviewers

We thank the Reviewers for their reviews and thoughtfulness during this process.

Reviewer 2

Comment 1: I thank the authors for providing exceptionally thorough and well-considered responses to my previous comments (as well as to those of the other reviewers). I believe this paper is a strong contribution to the literature and provides conceptual and methodological tools that will drive further innovation in emotion and other research domains.

Response 1: Thank you!

Reviewer 3:

Comment 1: I appreciate the authors' responses to my comments. I have no further comments. This manuscript is a fine addition the literature. Congrats!

Response 1: Thank you!